# An open source database for the synthesis of soil radiocarbon data: ISRaD version 1.0

Corey R. Lawrence[1][*][#], Jeffrey Beem-Miller[2#], Alison M. Hoyt[2,3#], Grey Monroe[4#], Carlos A. Sierra[2#], Shane Stoner[2], Katherine Heckman[5#], Joseph C. Blankinship[6#], Susan E. Crow[7#], Gavin McNicol[8], Susan Trumbore[2#], Paul A. Levine[9], Olga Vindušková[8], Katherine Todd-Brown[10], Craig Rasmussen[6#], Caitlin E. Hicks Pries[11#], Christina Schädel[12#], Karis McFarlane[13], Sebastian Doetterl[14], Christine Hatté[15], Yujie He[9], Claire Treat[16], Jennifer W. Harden[7,17], Margaret S. Torn[3], Cristian Estop-Aragonés[18], Asmeret Asefaw Berhe[19#], Marco Keiluweit[20#], Ágatha Della Rosa Kuhnen[2], Erika Marin-Spiotta[21#], Alain F. Plante[22#], Aaron Thompson[23#], Zheng Shi[24], Joshua P. Schimel[25#], Lydia J. S. Vaughn[3,26], Sophie F. von Fromm[2],

Rota Wagai[27#]

[1]US Geological Survey, Geoscience & Environmental Change Science Center, Denver, CO, USA

[2]Max Planck Institute for Biogeochemistry, Jena, Germany

[3]Climate and Ecosystem Sciences Division, Lawrence Berkeley National Laboratory, Berkeley, CA, USA

[4]Graduate Degree Program in Ecology, Colorado State University, Fort Collins, CO, USA

[5]US Forest Service Northern Research Station, Houghton, MI, USA

[6]Department of Soil, Water, and Environmental Science, University of Arizona, Tucson, AZ, USA

[7]University of Hawaii Manoa, Honolulu, HI, USA

[8]Department of Earth System Science, Stanford University, Stanford, CA, USA

[9]Department of Earth System Science, University of California, Irvine, CA, USA

[10]Wilfred Laurier University, Waterloo, Ontario, Canada

[11]Department of Biological Sciences, Dartmouth College, Hanover, NH, USA

[12]Center for Ecosystem Science and Society, Northern Arizona University, Flagstaff, AZ, USA

[13]Lawrence Livermore National Laboratory, Livermore, CA, USA

[14]Department of Environmental Systems Science, ETH, Zurich, Switzerland

[15]LSCE, UMR 8212 CEA-CNRS-UVSQ, Université Paris Saclay, F- 91191 Gif-sur-Yvette, France

[16]University of Eastern Finland, Joensuu, Finland

[17]US Geological Survey, Menlo Park, CA, USA

[18]Department of Renewable Resources, University of Alberta, Edmonton, Alberta, Canada

[19] Department of Life and Environmental Sciences, University of California, Merced, CA, USA

[20]Stockbridge School of Agriculture, University of Massachusetts-Amherst, Amherst, MA, USA

[21]Department of Geography, University of Wisconsin - Madison, Madison, WI, USA

[22]Department of Earth and Environmental Science, University of Pennsylvania, Philadelphia, PA, USA

[23]Department of Crop and Soil Science & Odum School of Ecology, University of Georgia, Athens, GA, USA

[24]Department of Ecology and Evolutionary Biology, University of California, Irvine, CA, USA

[25]Department of Ecology, Evolution, and Marine Biology, University of California, Santa Barbara, CA, USA

[26]Department of Integrative Biology, University of California, Berkeley, CA, USA

[27]National Agriculture and Food Research Organization, Institute for Agro-Environmental Sciences, Tsukuba, Ibaraki, Japan

**\*Corresponding Author (clawrence@usgs.gov), [#]USGS Powell Center working group participant**

**Abstract.** Radiocarbon is a critical constraint on our estimates of the timescales of soil carbon cycling that can aid in identifying mechanisms of carbon stabilization and destabilization, and improve forecast of soil carbon response to management or environmental change. Despite the wealth of soil radiocarbon data that has been reported over the past 75 years, the ability to apply these data to global scale questions is limited by our capacity to synthesize and compare measurements generated using a variety of methods. Here, we present the International Soil Radiocarbon Database (ISRaD, soilradiocarbon.org), an open-source archive of soils data that include reported measurements from bulk soils; distinct soil carbon pools isolated in the laboratory by a variety of soil fractionation methods; samples of soil gas or water collected interstitially from within an intact soil profile; $CO_2$ gas isolated from laboratory soil incubations; and fluxes collected *in situ* from a soil profile. The core of ISRaD is a relational database structured around individual datasets (entries) and organized hierarchically to report soil radiocarbon data, measured at different physical and temporal scales, as well as other soil or environmental properties that may also be measured and may assist with interpretation and context. Anyone may contribute their own data to the database by entering it into the ISRaD template and subjecting it to quality assurance protocols. ISRaD can be accessed through: (1) a web-based interface, (2) an R package (ISRaD), or (3) direct access to code and data through the GitHub repository, which hosts both code and data. The design of ISRaD allows for participants to become directly involved in the management, design, and application of ISRaD data. The synthesized dataset is available in two forms: the original data as reported by the authors of the datasets; and an enhanced dataset that includes ancillary geospatial data calculated within the ISRaD framework. ISRaD also provides data management tools in the ISRaD-R package that provide a starting point for data analysis; and as an open-source project, the broader soils community is invited and encouraged to add data, tools, and ideas for improvement. As a whole, ISRaD provides resources to aid our evaluation of soil dynamics across a range of spatial and temporal scales. The ISRaD v1.0 dataset (Lawrence et al., 2019) is archived and freely available at https://doi.org/10.5281/zenodo.2613911.

# 1. Introduction

The study of soil organic matter (SOM) dynamics is essential to an improved understanding of terrestrial ecosystem dynamics and the Earth's carbon cycle (Oades, 1988; Heimann and Reichstein, 2008). Current evaluations suggest that SOM accounts for up to 2770 Pg of organic carbon in the top 3 m of soil (Jackson et al., 2017; Le Quéré et al., 2018), which makes it one of the largest actively cycling terrestrial carbon reservoirs and an important modulator of climate change (Sulman et al., 2018). However, the lack of clarity about which fraction of that reservoir will respond to ongoing environmental changes (i.e. timescales of years to centuries) and which will respond only on millennial timescales (He et al. 2016) makes it imperative to improve our understanding of the controls on soil carbon cycling. Additionally, many studies and models focus on only the top 0.5 m of soil or less, despite deeper soils contributing a significant proportion of SOM storage by way of low carbon concentrations but large deep soil mass (Rumpel and Kögel-Knabner, 2010). There is an urgent need to synthesize a wide variety of soils data to model the role of soil in the climate system (Bradford et al., 2016), to develop more data-driven estimates of soil health (Harden et al., 2017), to inform policy and land management plans that preserve and enhance soil carbon storage (Minasny et al. 2017; Poulton et al. 2018), and to extend our detailed understanding of soil developed from observations made at the profile scale to both regional and global extents. Here we describe a new open-source database for the synthesis of soils data with a particular focus on soil radiocarbon data.

Radiocarbon (i.e., $^{14}C$) content of SOM is a useful tool to estimate the timescales of SOM cycling including the turnover time, residence time, or mean age of carbon in soil - defined as the time it has been isolated in soil from the atmosphere (Sierra et al., 2017; Manzoni et al., 2009; Trumbore, 2006). Although it was recognized very early on that radiocarbon measurements could provide a useful measure of the stability of soil carbon (Broecker and Olson 1960; Tam and Ostlund, 1960), the need for several grams of carbon for decay-counting methods meant that there were relatively few publications before the mid-1980's (e.g., Scharpenseel 1971; O'Brien & Stout, 1978). Many of these papers only published bulk soil radiocarbon for the same reason (with some exceptions, e.g., Martel and Paul, 1974; Goh et al. 1977). These early papers indicated that carbon in soils is heterogeneous and made up of a range of different aged materials that could be separated

chemically (Martel and Paul, 1974). Several of these studies use models of the uptake of bomb
carbon (Goh et al. 1977, Cherinski 1981, O'Brien, 1984; Balesdent, 1987). In the 1980's the
advent of accelerator mass spectrometry, a method that measures $^{14}C$ atoms in a sample by
accelerating them to high energy, allowed for radiocarbon analysis using milligrams of carbon
instead of grams, while simultaneously increasing sample throughput (Trumbore, 2009). This
development enabled analysis of small amounts of archived soils to track the incorporation of $^{14}C$
derived from atmospheric testing of nuclear weapons over time, as well as making it far easier to
analyze physically and chemically isolated soil fractions (e.g., Trumbore 1993). These applications
have led to an explosion in the number of publications with radiocarbon measurements from soil,
increasing from a few dozen papers annually during the 1980's to more than 150 per year in the
last decade (based on papers with "soil" and "radiocarbon" as keywords). The database presented
here is an attempt to provide an archive for all the previously published data but also a repository
for organizing new data as it is published.
Two recent soil radiocarbon synthesis efforts demonstrate the utility of these data for improving
predictions of SOM dynamics (He et al., 2016; Mathieu et al., 2015). Bulk soil radiocarbon
measurements, if not part of repeated time series, provide only an approximation of the time
elapsed since carbon in the soil was fixed from the atmosphere. In other words, soil carbon age as
measured by radiocarbon is defined as the age of carbon stored in the soil, since the time it enters
until a time of observation. However, this mean value is not representative of how fast soil carbon
will respond to a change in inputs, as it has been repeatedly demonstrated that SOM is not
homogeneous, and that carbon stabilized by different physical, chemical or biological mechanisms
cycles at different rates. Models can be used to explain time series of bulk radiocarbon or
physically and chemically separated SOM fractions, but this requires model structures with
multiple pools cycling on different timescales to simultaneously explain the rate of bomb $^{14}C$
uptake and the mean $^{14}C$ signature of SOM (Gaudinski et al. 2000; Baisden et al 2002a; 2013;
Sierra et al. 2012; Schrumpf et al. 2013). Partitioning SOM into pools is easily implemented in
models, but in reality, measuring these pools is both challenging and dependent on the techniques
used to fractionate the bulk soil (Moni et al., 2012) or to track throughput of bomb-derived carbon
through repeat measurements (Baisden et al., 2013; Baisden and Keller, 2013). A second measure
of carbon cycling rates in soils is the transit (residence) time, which is defined as the time it takes
carbon to pass through the soil system, since the time it enters until it is observed in an output flux
(Sierra et al., 2017). The modeled transit time can be constrained by measurements of the
radiocarbon signature of carbon in these output fluxes, which include respired $CO_2$ or dissolved
organic carbon (DOC) leached from the soil. Critically, most approaches using radiocarbon to
estimate the timescales of carbon cycling in soils require multiple measurements of carbon in
distinct soil reservoirs (Trumbore, 2000) or a time series of measurements made over the course
of several years (Baisden et al., 2013; Baisden and Keller, 2013). As the assumptions required for
modeling radiocarbon data can lead to confusion in the terminology and concepts of SOM
dynamics, it is imperative that we archive radiocarbon measurements in order to preserve the
ability to reevaluate calculations and compare data across different modeling frameworks.

Ongoing study of soils has led to shifting conceptual views of the controls on SOM dynamics
(Blankinship et al., 2018; Golchin et al., 1996; Lehmann and Kleber, 2015; Oades, 1989; Schmidt
and Torn et al., 2011). Current conceptual views that emphasize the protection of SOM from
microbial decomposition via physical isolation or sorption to soil mineral surfaces (Lehmann and
Kleber, 2015) and within anaerobic microsites (Keiluweit et al., 2016) have largely replaced earlier
paradigms of humification, selective preservation, and progressive decomposition. Three of the
fundamental questions currently driving SOM research are: (1) what are the controls on the
partitioning of organic inputs between soil reservoirs cycling over different timescales; (2) what
factors determine rates at which SOM in each reservoir is lost, retained, or transferred within the
soil; and (3) which mechanisms contribute to transformation of SOM to stabilized or more
protected forms? To address these questions, researchers typically measure the concentration or
mass content of organic carbon along with other properties, including molecular composition,
isotopic ratios, and the distribution of SOM between conceptually or operationally defined pools
(e.g., Basile-Doelsch et al., 2009) or a time series of samples collected over the course of decades
(e.g., Baisden et al, 2002a).

Soil fractionation is the operationally defined separation of soils into distinct pools or "fractions"
through a variety of physical, chemical, and biological approaches. Soil fractionation is generally
intended to isolate soil fractions that reflect SOM in different physico-chemical states or
mechanisms of SOM protection (Trumbore and Zheng, 1996); these mechanisms may operate on
distinct temporal scales (e.g., Khomo et al., 2017). For example, density fractionation of SOM is
a commonly applied technique (Golchin et al., 1994; 1995; Crow et al., 2007; Sollins et al., 2006;
2009; Swanston et al. 2005). The "light" fraction of soil material that floats in a dense solution
(e.g., sodium polytungstate) or gets picked up by electrostatic attraction (Kaiser et al., 2009) is
sometimes used as a proxy for rapidly-cycling SOM, as this material is generally observed to have
a shorter mean residence time compared with the bulk soil average, while the "heavy" or dense
material is used as a proxy for mineral-associated SOM, which is assumed to cycle more slowly
(e.g., Sollins et al., 2009). In some cases, sonication of the suspension may be used to further
isolate occluded SOM, i.e., organic material in soil aggregates (Golchin et al., 1994; Kasier and
Berhe, 2014). Other methods for isolating SOM with different cycling rates in the soil include, but
are not limited to, physical separation of aggregates by size and water-stability (Jastrow et al.,
2006; Plante et al., 2006; Six and Paustian, 2014) or of different-sized soil particles (Desjardins et
al., 1994), biological incubation of soils (Torn et al., 2005; Trumbore, 2000; Paul, et al., 2001),
and chemical extractions (Heckman et al., 2018; Masiello et al., 2004).

Comparing the mass and radiocarbon signature of the carbon leaving or entering the soil system
(fluxes) with those of specific soil fractions provides insight into the rates of transfers between
pools and provides a means for differentiating between various measures of dynamics ranging
from mean age to the transit time of carbon for the whole soil, a given depth increment, or a given
SOM pool (Gaudinski et al., 2000; Baisden et al., 2002a; 2002b; 2003; Sierra et al., 2014; Ohno
et al., 2017; Ziegler et al., 2017; Szymanski et al., 2019). Similarly, measurements of interstitial
soil carbon (i.e., in soil water or gases collected from within an intact soil profile) and its isotopic
signature provide key information about the dynamics of the carbon present in the soil solution
(Sanderman et al., 2008). Soluble carbon is believed to be the dominant pathway for vertical
transport of organic carbon (Kaiser and Kalbitz, 2012; Angst et al., 2016), and also an intermediate
stage through which carbon exchanges from being vulnerable to microbial decomposition to being
stabilized on mineral surfaces (Jackson et al., 2017; Leinemann et al., 2018).

Measurements of bulk soils as well as soil fractions are evaluated in the context of other soil
properties to better understand the controls on SOM preservation. However, the diversity of soil
fractionation methods makes it difficult to compare measurements across soils or to evaluate best
practices (e.g., Trumbore and Zheng, 1996). Combining radiocarbon measurements of soil carbon
fractions, time series, incubations, interstitial observations, and fluxes has proven useful in
resolving the contribution of different soil carbon persistence mechanisms in a site-specific
modeling context (Braakhekke et al., 2015), but the application of this approach beyond the site-
scale has thus far been limited due to the lack of globally synthesized data.

With a changing paradigm for SOM dynamics and ever-evolving SOM models, it is more
important now than ever that we synthesize existing soil radiocarbon measurements *and* provide a
central repository for new data. There have been previous efforts to develop a soil radiocarbon
database (Becker-Heidmann, 1996; 2010; Trumbore et al. 2011), separately or integrated with a
general-purpose soil carbon database (Harden et al., 2017). However, a challenge remains: to
compile and organize soil radiocarbon data that has been collected in many different and complex
ways (e.g., using various fractionation methods or including fluxes as well as organic matter
pools). Addressing this challenge will provide new opportunities to leverage existing soil
radiocarbon data for critical research such as developing practical and theoretical insights into the
information contained in various fractionation methods and how they relate to one another. This
will expand our understanding of controls on soil carbon dynamics, and facilitate broader
integration of radiocarbon constraints on soil carbon turnover in Earth system models. For
example, He et al. (2016) leveraged a synthesis of bulk-soil radiocarbon data to better constrain
the age of carbon in five Earth system models, demonstrating that without this added constraint,
these models overestimate soil carbon sequestration potential by an average of 40%.

Here, we present a flexible database spanning broad spatial scales and capturing a range of data
types including diverse soil fractionation methods, incubations, fluxes, interstitial measurements
and spanning a range of spatial scales. Our goal is to provide an open-access data resource that
will encourage the scientific community to apply the database for a variety of synthesis studies or
metaanalyses, and also contribute data to the repository.

## 2. The International Soil Radiocarbon Database (ISRaD)

The International Soil Radiocarbon Database (ISRaD) is designed to be an open-source platform that (1) provides a repository for soil radiocarbon and associated measurements, (2) is able to accommodate data collected from a large variety of soil radiocarbon studies, including the diversity of fractionation techniques applied to soils as well as repeated bulk measurements made over spatial or temporal gradients, and (3) is flexible and adaptable enough to accommodate new variables and data types. Although ISRaD was specifically developed with soil radiocarbon measurements in mind, it is well suited for synthesizing other soil measurements, including stable carbon and nitrogen isotopes. Importantly, we currently focus only on natural abundance isotopic measurements and therefore exclude data from isotopic tracer studies. The ISRaD v1.x data is archived and freely available at https://doi.org/10.5281/zenodo.2613911 (Lawrence et al., 2019). Access to additional information as well as the various ISRaD resources described below is provided through the ISRaD website (soilradiocarbon.org).

### 2.1 Database and Dataset Structure

In its most general form, ISRaD is an implicitly relational database. It consists of a linked hierarchical list of tables that contain soil measurements, i.e. variables (Fig. 1). The fundamental unit of organization in ISRaD is the *entry*, which corresponds to a unique dataset i.e., a dataset with a digital object identifier (DOI), while each subordinate table corresponds to data from that entry with a particular spatial or temporal dimension.

Transparency and traceability are fundamental tenants of ISRaD. Accordingly, each entry, whether ingested individually or as a compilation, must have a DOI. For data from published studies, the DOI of the publication is acceptable. Data from unpublished studies must be registered for a DOI through a DOI registration agency (e.g., zenodo.org, www.pangaea.de, etc.) prior to ingestion into ISRaD. As it is equally important to be able to reconstruct prior data compilations e.g., synthesis studies, the specific references for individual datasets making up a synthesis are ingested as part of the synthesis entry and the entry is flagged within the database with an additional reference to the synthesis study itself. For example, several of the major data sources added to ISRaD were synthesis studies (e.g., He et al., 2016; Mathieu et al., 2015), and users can generate reports of data from these prior syntheses by constructing a query that utilizes this synthesis flag.


Each ISRaD release will be available in two forms: (1) a raw version of data (*ISRaD_data*),
containing only values that were reported in the original source of each data entry and (2) an
expanded version of the database (*ISRaD_extra*). The *ISRaD_extra* version of the dataset includes
additional parameters that have either been calculated or imported based on site coordinates, such
as geospatially referenced climate information. Table 2 includes examples of some of the new
variables included in *ISRaD_extra*; a more detailed list of this growing list of variables can be
found on the ISRaD website in the ISRaD_extra Information File. Both versions of the database
follow the general data hierarchy outlined below.

2.2 Data Hierarchy
The ISRaD data hierarchy consists of eight levels of information (Fig. 2). The top level of the data
hierarchy is the metadata table (1), which includes information describing the source of data for a
particular entry. The remainder of the hierarchical levels can be defined by the spatial extent of the
information included in each table. The site (2), profile (3), layer (4), and fraction (5) tables
represent information captured from decreasing spatial extents: from the scale of the study area to
individual mass fractions isolated from a single soil sample. Special cases of the last three spatial
extents further accommodate the temporal context of repeated measurements: (6) fluxes, (7)
interstitial, and (8) incubations. In the sub-sections below, we provide overviews and examples of
the types of information reported at each level, and for each of the tables that occupy these levels
(Fig. 2).

The data hierarchy is maintained across tables through the use of unique keys, or linking variables
(noted with a "*" in the following descriptions) that are required in each record (row) of data in
each table. In addition to the table-specific key, each subordinate table in the hierarchy must also
contain the key variables of the above tables. For example, in addition to a unique *layer_name\**,
each record in the layer table must also be associated with an *entry_name\**, *site_name\**, and
*pro_name\** (profile name) i.e., the key variables for the metadata, site, and profile tables.

ISRaD provides basic quality assurance/quality control (QA/QC) protocols (described below) and
expert review that are applied prior to ingesting entries. These protocols are used to ensure required
variables are complete, that the key variables match across levels of the hierarchy (more detail
below), and data entered match the specified data type and range for a given variable. Variables
that are not designated as required need only be completed if those data are available. The ISRaD
template and a detailed description containing the full list of variables along with instructions for
populating the template can be downloaded or viewed from the "Contribute" page of the web-
interface (soilradiocarbon.org).
For all variables across all hierarchical levels, it is important to observe the acceptable data types
(character, numeric) and units. Variable names, descriptions, and reporting conventions are given
in the heading columns of the ISRaD template file (ISRaD_Template.xlsx) and more detailed
information is provided in the data dictionary (ISRaD_Template_Info.xlsx). Allowed values
include unrestricted text, controlled text, or numeric variables with or without defined ranges.
Unrestricted text is generally limited to naming and note data fields, while controlled text fields
are implemented for certain variables in an attempt to standardize the data and simplify data
analysis. In the event that desired variables are not included in the current version of ISRaD, users
may submit a request to add new variables. This process is initiated by posting an issue at the
ISRaD GitHub repository and is described in more detail in section 3.4.

2.2.1 Metadata Table
The metadata table provides information for the characterization of the entry itself. Required
metadata includes the entry name (i.e., *entry_name\**), the DOI, the data curator (the person who
oversees template entry), and their contact information. The entry name is the key variable used to
match the entry with measurements reported at the other data levels.

2.2.2 Site Data Table
Site-level data are limited to the geospatial details defining the coarsest scale of the study area(s)
included in each entry. We define a site as a spatially defined location that includes one or more
soil profiles. By convention, we define a site as having $\geq 5$ km radius, i.e., samples collected within
5 km of each other should be grouped under the same "site" designation. However, the 5 km radius
is a convention only, as the distinction between site and profile may be study-specific, and
geospatial data at this resolution is not always available for legacy datasets. Spatial coordinates are
required to designate a site, and thus the required fields at the site level are limited to the site name
(*site_name\**), latitude, and longitude. Every entry must specify a minimum of one site location,
but can include multiple sites that do not need to be located in close proximity. For entries that do
not report spatial coordinates, the data curator may estimate latitude and longitude based on the
description of the study area using any of the widely acceptable mapping software (e.g., Google
Earth, Google Maps, etc.). The site table does not include fields for reporting site properties. Such
directly measured variables are reported at the profile level. The intended purpose of the site level
data is to provide at least coarse-scale geospatial coordinates for extracting consistently sourced
parameters from geospatial datasets, which can then be compared against the range of
measurements reported at the profile level.

### 2.2.3 Profile Data Table
Profile-level data includes details pertaining to specific sampling locations. If available, profile-
scale spatial coordinates should be provided in addition to site-scale coordinates.

Many variables that may initially appear to belong at the site level are instead included at the
profile level to facilitate accurate representation of spatial heterogeneity at a finer scale than the
site level (e.g., for multiple profiles observed at the same site). Examples include local mean annual
temperature and precipitation, soil taxonomic classification, vegetation type, land cover, depth to
bedrock, and parent material composition. Other than the entry name and site name, the only
additional required variable at the profile-level is the profile name (*pro_name\**).

### 2.2.4 Flux Data Table
Soil flux data present a special case of observations that correspond to the profile level of the
database hierarchy. Flux-level data allows for reporting of temporally explicit measurements of
mass or energy transfer occurring at the profile scale. Both gas and liquid analytes (e.g. $CO_2$, $CH_4$,
dissolved OC, particulate OC, etc.) may be reported in flux data. In addition to the profile name,
records with flux data must also include the observation date (*flx_obs_date*). Data measured at
multiple time points in a single location will have identical profile names but unique temporal data.

2.2.5 Layer Data Table
Layer-level data corresponds to measurements made for a specific depth increment collected from
a soil profile. The required variables at the layer level include layer name (*lyr_name*), depth of
layer top, and depth of layer bottom. The latter two variables describe the upper and lower range
of the sampling depth, respectively, in units of centimeters. We use a depth reporting system where
the top of the mineral soil is denoted as zero and subsequent depths below that point are reported
with incrementally increasing positive values. Organic horizons are thus reported as negative depth
intervals. Special indicator fields (e.g., *lyr_all_org_neg*) are used when the depth to the mineral
soil is unknown, e.g. for deep organic horizons or peats. The layer level is where most common
measurements of soil physical, chemical, and/or biological properties are reported. As such, there
is an ever-increasing list of variables that may be reported in the layer table. Users should consult
the up-to-date template instruction file for the complete list of accepted variables.

2.2.6 Interstitial Data Table
The interstitial level is a special case of layer-level data. Specifically, interstitial data refers to
measurements made on material occupying the interstices of the soil structure. In most cases, this
material can be thought of as being mobile relative to the rest of the soil matter. Some common
examples include gases, liquids, and colloids. Like flux data, the interstitial data table
accommodates repeated measurements of these properties through time and as such, the
observation date must be recorded for each record in the interstitial table. Because interstitial
records may not correspond to the same depth increments defined for solid phase analyses, separate
depth reporting is used in the interstitial table distinct from what is reported in the layer table. Both
sampling methodology as well as the properties of interstitial samples are reported in the interstitial
table.

2.2.7 Fraction Data Table
Compared with most other soil databases, the fraction data table of ISRaD is unique. The fraction
data fields are designed to accommodate and allow for fair comparison of the wide-ranging
methodologies utilized to partition soils into discrete fractions. As such, there are more required
fields for the fraction level compared with the other hierarchical levels. These required fields
include fraction name (*frc_name\**); the input source (*frc_input*), which can be the name of another
fraction or bulk (unfractionated) soil layer fraction scheme (*frc_scheme*), which is a controlled set
of terms describing the general class of fractionation procedure used; the fractionation agent
(*frc_agent*), which provides additional detail for methods that have multiple options; the upper and
lower boundaries (*frc_lower* and *frc_upper*), which allow for description of the fractionation
thresholds used in the fractionation procedure; and finally the fraction scheme units
(*frc_scheme_units*), which describes the units of reference for the cut-off thresholds.

For example (see figure 3), most soil density fraction (*frc_scheme* = density) procedures starts with
bulk soil from the layer in question (*frc_input* = bulk). The first distinct fraction, "free light", is
isolated by floating the soil in a heavy solution, e.g. sodium polytungstate (frc_agent = SPT). If
the density of the sodium polytungstate used in density separation step was 1.6 g cm$^{-3}$, *frc_lower*
for the "free light" fraction = 0, and *frc_upper* = 1.6 (indicating that anything with a density less
than 1.6 was included), and *frc_scheme_units* = g cm$^3$. In addition to these required fields, the
fraction-level data may include many of the same data fields that are reported for the layer-level
data. Ideally the fraction data also includes the mass percentage of the total sample represented by
the fraction as well as the specific carbon concentration and carbon isotopic composition of the
fraction, which are critical for relating bulk and fraction level observations.

2.2.8 Incubation Data Table
Flux rates and isotopic signatures of laboratory-incubated samples are reported in the incubation
table. Sample processing data (e.g., whether or not roots have been removed from samples prior
to incubation) are recorded, as well as incubation conditions (e.g. temperature, moisture, duration).
Repeat measurements, such as incubation time series, can also be recorded. Incubation records
must be linked either to a layer or both a fraction and a layer, e.g. roots isolated from a specific
bulk layer sample.

2.3 Radiocarbon Data – Reporting Conventions
Radiocarbon measurements of environmental samples have a long history, much of which is
reviewed in Trumbore (2009) including common units. Radiocarbon data ingested to ISRaD are
required to adhere to some basic reporting conventions. First, measurements of radiocarbon may
be reported in units of either fraction modern (FM) or $\Delta^{14}$C. Within the *ISRaD_extra* version of
the database, values reported in one or the other accepted units are internally converted and filled
across all entries, so that either unit may be used for analyses of the full dataset. Other units are
not supported at this time – for example, calibrated radiocarbon dates are not accepted, as the
calibration curves are evolving over time. Such calibrated ages make sense only for certain
fractions (e.g. macrofossils found in soils), and do not make sense in the context of most soil
organic matter, which is an open system for carbon, with inputs that vary in $\Delta^{14}C$ over time. For
datasets where radiocarbon is reported in units other that FM or $\Delta^{14}C$ (e.g. percent modern carbon
or conventional radiocarbon age), it is up to the data curator or original author of the dataset to
convert the reported values to one or both of the permitted units. Second, the year of measurement
for each radiocarbon value must also be reported so that values may be internally converted
between the two accepted units. In addition to these basic requirements, there are several other
optional fields pertaining to radiocarbon data. These include the radiocarbon laboratory; the
laboratory number, a unique identifier issued by each AMS facility; the analytical error reported
for each measurement reported by most laboratories; and the environmental standard deviation of
replicate samples (if analyzed). These variables are not required for data submission but should be
included if they are available.

2.4 Data Ingestion
New data entries are added, or ingested, into ISRaD through a user-initiated process. The most
common means of ingesting entries is via the template provided on the ISRaD website
(*ISRaD_Master_Template.xlsx*). The template is intended to be used in combination with the data
dictionary (*ISRaD_Template_Info.xlsx*). These files and other supporting documentation (user
guide and FAQ) are also available at the website (soilradiocarbon.org). Completed entries that
have been formatted for ingestion must also pass the automated QA/QC test before the ingestion
process can proceed. Users can initiate QA/QC using the ISRaD-R package (described below), or
directly from the ISRaD website. If the entry fails QA/QC, the report from the test can be used as
a guide to make corrections. Once an entry passes QA/QC, it can be submitted for the final two
steps of the ingestion process: expert review and final ingestion.

Data templates that have passed QA/QC should be submitted via email to ISRaD at
info.israd@gmail.com. These templates are then distributed to ISRaD expert reviewers who
inspect template files to ensure proper completion of the more complex aspects of the template,
such as classification of soil fractionation methods. If problems are identified with a submitted
dataset during the expert review process, reviewers will work with the data curator to ensure these
problems are corrected. Once the expert reviewer signs off on a submitted template, it will be
ingested into the database.

## 3. Database Infrastructure

### 3.1 ISRaD v1.x

The current version, v1.4, of ISRaD includes a total of 212 individual data entries and 550 sites
spanning the globe (Fig. 3). The current distribution of data across the various levels of the
database hierarchy are shown in Table 1, and a full list of data entry references is provided in
Supplemental Table 1.

Users may access ISRaD and its supporting information three ways: (1) the website, (2) the ISRaD-
R package, and (3) the GitHub repository. Each of these access points is described in more detail
below.

### 3.2 The ISRaD Website

Most simply, users can access ISRaD data and associated resources by way of the website
(http://soilradiocarbon.org). From the website users can download pre-compiled versions of the
database in a simple file format ((.xlsx or .csv), which can be easily ingested into graphical or
database software. The website also provides access to the most recent versions of the ISRaD entry
template, the template information file, an up-to-date list of the datasets included in the latest
version of ISRaD, the QA/QC tool, and a variety of other resources for assisting with filling out
data templates and interacting with ISRaD data.

### 3.3 Accessing ISRaD within the R computing environment and the ISRaD-R package

ISRaD has been designed for ease of use in the R computing environment (R Foundation for
Statistical Computing, Vienna, Austria) in order for users to be able to take advantage of the full
suite of R capabilities and functionality to manipulate and analyze ISRaD data.  Many of the basic
functionalities such as loading current versions of the ISRaD data objects can be performed in the
R environment without installation of the ISRaD-R package. A number of vignettes including R
scripts for some commonly used data manipulations or plotting are given on the website.
Users who need to locally compile a version of ISRaD (e.g. using their own templates) or who
want access to the full suite of reporting functions can access these features by installing the
ISRaD-R package (also called *ISRaD*, which is available at the CRAN repository, http://cran.r-
project.org/). The ISRaD-R package primarily consists of the code that is used to assemble the
database and perform the QA/QC checks for the ISRaD datasets. Users may ingest their own data
within a local version of the ISRaD dataset by running the function ISRaD::compile(). This
functionality is intended to allow researchers to interpret their own new or unpublished data in the
context of ISRaD data. Additionally, the ISRaD-R package provides simple tools to download and
quickly import the most recent ISRaD data into the R environment, and produce basic data
summaries and visualizations for the full dataset or user-defined subsets using report functions.
3.4 The ISRaD GitHub Repository
The source code for the ISRaD-R package is hosted under version control on the GitHub repository
ISRaD (https://github.com/International-Soil-Radiocarbon-Database/ISRaD) (GitHub Inc., San
Francisco, CA). This platform is used to facilitate the open-source collaborative development of
ISRaD data and additional database tools. Through the GitHub interface, users may (1) access data
entries included in the compiled database; (2) evaluate and suggest modifications of the underlying
code used for compilation, QA/QC, and calculation of the additional variables included in either
the compiled ISRaD data object (*ISRaD_data*) or in the augmented data product (*ISRaD_extra*);
and (3) report problems, questions, or other issues.
3.5 The Soil Carbon Information Hub
For students or non-experts interested in learning more about the science behind the data, we have
developed the Soil Organic Carbon Information Hub (SOC-Hub). SOC-Hub (https://international-
soil-radiocarbon-database.github.io/SOC-Hub/) is a set of articles in the form of blog posts
providing background information on soils, radiocarbon, the terrestrial carbon cycle, and soil
models. A large portion of the content for this site was created by students, and whenever possible
uses non-technical language to describe topics pertinent to ISRaD. Technical editing of the SOC-
Hub is facilitated through the ISRaD GitHub repository. Users are welcome and encouraged to
contribute to or improve the content in SOC-Hub. We will update the SOC-Hub annually.

# 4. Database Operations


4.1 Accessing data entries
Individual data entries (i.e., completed templates, or templates output from ingested compilations)
that have passed QA/QC and the expert review process are hosted in the "ISRaD_data_files" folder
of the GitHub repository. Users may download these entries in order to add new data or to make
corrections to existing data if problems are discovered. Corrected files can be resubmitted to
ISRaD once they pass QA/QC by emailing the updated template and a text file of the QA/QC
report to the ISRaD editor (info.israd@gmail.com), and will be reingested after passing the expert
review process. This process of user-initiated revision of existing data entries is particularly useful
when large data compilations are ingested into ISRaD from previously published syntheses (e.g.,
He et al., 2016; Mathieu et al., 2015) or when publications report treatment means. Depending on
the scope of the synthesis efforts, entries ingested into ISRaD this way may omit data available
from the original studies, and the entry modification process allows those data to be added or
corrected as needed.

4.2 Accessing code
Access to the source code underlying the ISRaD database compilation and calculations allows for
users to check for errors and contribute to the functionality of ISRaD. Users with a registered
GitHub account are invited to write code that adds to or improves upon the existing database tools.
Using standard GitHub tools, users will submit a "pull request", and following code testing and
evaluation of utility to the ISRaD community, user-submitted code will be incorporated into the
ISRaD-R package.

4.3 Reporting Issues, making suggestions, and asking questions.
One of the most important tools available to ISRaD users is the ability to post questions, report
issues, or make suggestions, including requests to incorporate new variables. We use issue tracking
tools provided by GitHub to track and categorize user input including: suggestions for
improvements; problems or errors with website, the R-package, code, or any other aspects of
ISRaD; requests for new variables or issues related to existing variables (e.g., incorrect acceptable
ranges used in QA/QC); or asking questions related to template entry or any other aspect of ISRaD.
While the GitHub issue-reporting functionality is the preferred means for reporting questions or
issues with the database or process, it does require that users register a GitHub account. Users who
do not wish to or are not able to register with GitHub, can also submit issues or questions via an
email to the ISRaD editor (info.israd@gmail.com), however, the response time may be slower.

4.4 Database Versioning and Archiving
Versioning of ISRaD datasets will be tracked on two levels: official releases and regular
updates. Official releases of the ISRaD datasets (e.g., ISRaD v1.0) will be issued
periodically, following major changes to the codebase or after the ingestion substantial new
data. For each official release, a DOI number will be issued and the data will be archived
at Zenodo (https://zenodo.org/) and at the USGS Science Base repository. These archives
will be maintained into perpetuity to facilitate reproduction of any analyses conducted
using a past version of the database. Regular updates correspond to minor changes in
development version of ISRaD. These regular updates occur anytime the codebase is
rebuilt or when new data is ingested. Regular updates will be archived and available
through the GitHub repository, but they will not be issued DOI numbers. The most current
version of the ISRaD datasets are always available from the ISRaD website and the GitHub
repository. To ensure repeatability of analyses and accurate citations, users accessing the
ISRaD dataset should always record version number of the data. The names of ISRaD data
files reflect the dataset versioning using the following standardized structure: (*data
name*)(*vX*)(*date*).(*format*), where *data name* reflects the file type and structure (e.g.,
ISRaD_extra_flat_layer), *vX* refers to the most recent official release version number
(X), *date* corresponds to date of the most recent regular update of the database in the
GitHub repository, and *format* is the file type.

Versioning of the ISRaD codebase and the ISRaD-R package are tracked separately from ISRaD
data. The codebase versioning is tracked via Git but can be linked back to the data version using
the *date* identifier from the data file names. The ISRaD-R package versioning is tracked via the
Comprehensive R Archive Network (CRAN) and is independent of the ISRaD data versioning.
Thus, when using the ISRaD-R package it is useful to record the version installed from CRAN
but users must also remember to record which version of the data they have accessed.

4.5 Citing ISRaD
Users citing ISRaD should cite this publication as well as the most recent official data release at
the time that they accessed the data. In their citation of the official release, users should also
reference the version of the data they used (e.g., v1-2019-09-09).

4.6 Data Sharing Between Soil Databases
ISRaD is not the only soils database available to the international research community. The
primary niche of ISRaD is the ability to synthesize soil radiocarbon data and provide a framework
for comparing soil carbon fraction data. For other purposes, there may be other soil databases that
are more applicable. However, as a benefit of adding data to ISRaD, we facilitate sharing of data
ingested into ISRaD with other databases developed by the soil science community. At present,
ISRaD has a reciprocal agreement with the International Soil Carbon Network (ISCN), which is
focused on soil carbon content and related variables from bulk soils (i.e., no isotope or soil fraction
information). As per this agreement, the ISCN retrieves bulk soil data from ISRaD, and is
responsible for filtering duplicate entries and incorporating any new data into the ISCN database.
# 5. Database Governance
ISRaD is a community effort with multiple contributors operating at different levels. Governance
of ISRaD is required in order to ensure continuity of services and to plan for the future evolution
of this data repository. The governance structure of the ISRaD is pyramid shaped (Fig. 5). The
ISRaD *scientific steering committee* (SSC) consists of a rotating group of 7 scientists and data
managers. The committee members are nominated and voted into service by a majority vote of the
existing steering committee. The role of the steering committee is to determine the feasibility of
major changes to ISRaD proposed by the community; to oversee data management, archiving, and
establishment of cooperative agreements; and to coordinate activities and funding of affiliated
institutions.

Database *maintainers* oversee the development and maintenance of the technical resources
underlying ISRaD. For example, these individuals are responsible for overseeing GitHub pull
requests and managing major changes in the ISRaD data template and/or data structure. The ISRaD
associate editor is a special case of maintainer, whose role also includes assigning submitted
templates to expert reviewers (described below) and periodically rebuilding the database with new
entries that have passed the expert review process.

Data *contributors* are users who contribute data to ISRaD. Anyone can be a data contributor
provided they agree to the terms of use and follow the proper steps for contributing data to ISRaD.
Within the pool of data contributors, individuals with significant experience working within the
ISRaD structure may be designated, either by the steering committee or database maintainers, as
*expert reviewers*. These individuals are tasked to assist maintainers and oversee peer review of
contributed entries. Although the automated QA/QC tools are designed to catch many common
errors in the data ingestion process, review by these expert contributors ensures the integrity of the
data within ISRaD.

Finally, ISRaD *data users* are individuals who are accessing ISRaD or ISRaD-supported resources
to utilize data and other resources rather than to contribute data. Anyone can be a data user
provided they agree to the basic user guidelines and terms of use described in the next section.

Although the structure of the ISRaD governance pyramid is oriented around individual users, the
nature of scientific research is often more group-focused. For example, teams of researchers
generally work together to seek out funding and to conduct research. Thus, in some cases a group
or team of individuals may seek to utilize or modify ISRaD for their purposes. Such groups can
petition the scientific steering committee to be formally designated as an ISRaD organization. This
process should be followed when groups seek to leverage the ISRaD resources beyond the scope
of a basic user or contributor. The steering committee will consider the scope of the work proposed
by the group and, when appropriate, provide a letter of support for funding proposals. Approved
organizations should nominate a member to serve on the steering committee and, in the case of
organizations making large changes or additions to ISRaD, a data maintainer to coordinate the
technical aspects of that work.

## 6. Database Availability and User Guidelines

As detailed above, ISRaD is an open source project that provides several ways for participation.
ISRaD v1.0 data (Lawrence et al., 2019) is archived and freely available at
https://doi.org/10.5281/zenodo.2613911 Anyone may share or adapt the ISRaD dataset provided
they do so in accordance with the Creative Commons Attribution 4.0 International Public License
(https://creativecommons.org/licenses/by/4.0/legalcode), also referred to as CC-By. In addition,
we strongly encourage ISRaD users to follow two simple guidelines for use:

(1) When utilizing the resources provided by ISRaD, including the complete dataset,
individually curated entries, or value-added calculations included in the R-package, users
should cite this publication and reference the version of ISRaD that was used for their work
(see section 3.6 above). Additionally, if users leverage individual data entries from the
database, they should also cite the original source dataset and/or paper.
(2) When users interpret their own data in the context of data accessed from ISRaD, they
should submit those new data for inclusion in ISRaD after they have published their results
and/or obtained a DOI for their dataset.

## 7. Conclusions and Outlook

ISRaD is an interactive open source data repository specializing in radiocarbon data associated
with measurements of soils spanning a broad range of spatial scales. The ISRaD dataset is unique
in that it includes not only measurements of bulk soils but also measurements of soil water, gases,
and the wide diversity of soil pools isolated through different fractionation methodologies. Most
of the studies summarized in ISRaD were conducted with a goal of understanding the factors
controlling timescales of carbon cycling in specific sites, regions or biomes. ISRaD is an attempt
to gather the data from these individual studies in one place and in the same format to facilitate
comparisons and synthesis activities. There are three ways through which potential users can
access ISRaD: (1) the web-interface enables users to download of the most recently compiled
report formatted as a .csv file, (2) the ISRaD-R package provides access to the compiled reports

as well as visualization tools and R-based querying tools, or (3) the GitHub repository provides direct access to the source code for the ISRaD-R package, as well as data from individual entries and the compiled database. Currently, the ISRaD dataset contains ~8500 radiocarbon analyses, which, at a typical cost ~$500 each, represent over US $4,250,000 dollars of research investment. By providing a useful platform for existing data, we hope to encourage the community to increase the effectiveness of that investment, and to use the ISRaD platform as a repository to increase the impact of new results. Many opportunities exist for applying ISRaD data for improving our understanding of controls on soil carbon dynamics, for comparing different methodologies of characterizing soils, or for constraining soil processes in models ranging from profile to global scales.

## 8. Author Contribution

The creation of ISRaD was a community effort. The initial concept to build ISRaD started with the USGS Powell Center working group on Soil Carbon Storage and Stability but benefited greatly from early efforts of the International Soil Carbon Network and other individual efforts to compile soil fraction or radiocarbon data. Scientists from the Max Planck Institute for Biogeochemistry joined forces with the Powell Center group to greatly expand the scope and technical complexity of ISRaD. CL, JBM, AH, GM, CS, SS, KH, JB, SC, GMc, and ST designed and built ISRaD as well as led the preparation of the manuscript. PL, OV, KTB, CR, CHP, CS, KM, and SD provided technical contributions, including coding, to the creation of the database as well as assisted with the ingestion of data. CH, YH, CT, JH, MT, and CEA provided large datasets or data compilations. AAB, MK, AK, EMS, AP, AT, JS, LV, SFvF and RW contributed to the conceptual framing of ISRaD and assisted with data ingestion. All authors read and commented on the manuscript.

## 9. Acknowledgments

We gratefully acknowledge funding support for this work from the US Geological Survey Powell Center for the working group on Soil Carbon Storage, the Max Planck Institute for Biogeochemistry, the European Research Council (Horizon 2020 research and innovation programme - grant agreement No. 695101), the USGS Land Change Science mission area, and the US Department of Agriculture (Soil Carbon Working Group award #2018-67003-27935). Troy Baisden, Jonathan Sanderman, Lisamarie Windham-Myers, Sanjay Advani and several

606    anonymous reviewers provided constructive reviews, which improved the quality of this work.

607    Any use of trade, firm, or product names is for descriptive purposes only and does not imply

608    endorsement by the U.S. Government.

# 10. Citations

Anderson, D.W. and Paul, E.A. Organo-mineral complexes and their study by radiocarbon dating. Soil Science Society of America Journal, 48(2), 298-301, 1984.

Angst, G., Kögel-Knabner, I., Kirfel, K., Hertel, D. and Mueller, C.W.: Spatial distribution and chemical composition of soil organic matter fractions in rhizosphere and non-rhizosphere soil under European beech (Fagus sylvatica L.), Geoderma, 264, 179–187, doi:10.1016/j.geoderma.2015.10.016, 2016.

Balesdent, J. The turnover of soil organic fractions estimated by radiocarbon dating. Science of the Total Environment, 62, 405-408, 1987.

Baisden, W.T., Amundson, R., Brenner, D.L., Cook, A.C., Kendall, C., Harden, J. A Multi-Isotope C and N Modeling Analysis of Soil Organic Matter Turnover and Transport as a Function of Soil Depth in a California Annual Grassland Soil Chronosequence. Global Biogeochemical Cycles 16((4)): 1135, doi: 1029/2001GB001823. 2002a.

Baisden, W.T., Amundson, R., Cook, A.C., Brenner, D.L. The turnover and storage of C and N in five density fractions from California annual grassland surface soil. Global Biogeochemical Cycles 16: doi: 1029/2001GB001822. 2002b.

Baisden, W.T., Amundson, R. An analytical approach to ecosystem biogeochemistry modeling. Ecological Applications 13(3): 649-663. 2003.

Baisden, W.T., Parfitt, R.L., Ross, C., Schipper, L. A. and Canessa, S.: Evaluating 50 years of time-series soil radiocarbon data: towards routine calculation of robust C residence times, Biogeochemistry, 112(1-3), 129–137, doi:10.1007/s10533-011-9675-y, 2013.

Baisden, W.T. & Keller, E.D. Synthetic Constraint of Soil C dynamics Using 50 Years of Radiocarbon and Net Primary Production (NPP) in a New Zealand Grassland Site. Radiocarbon 55(2-3): 1071-1076. 2013.

Basile-Doelsch, I., Brun, T., Borschneck, D., Masion, A., Marol, C. and Balesdent, J.: Effect of landuse on organic matter stabilized in organomineral complexes: A study combining density fractionation, mineralogy and delta C-13, Geoderma, 151(3-4), 77–86, doi:10.1016/j.geoderma.2009.03.008, 2009.

Becker-Heidmann, P. Requirements for An International Radiocarbon Soils Database, Radiocarbon, 38(02), 177–180, doi:10.1017/S0033822200017549, 1996.

Becker-Heidmann, P. A new attempt to establish the international radiocarbon soils database (IRSDB), Radiocarbon, 52, 1405–1410, 2010.

Blankinship, J.C., Berhe, A.A., Crow, S. E., Druhan, J.L., Heckman, K.A., Keiluweit, M., Lawrence, C.R., Marin-Spiotta, E., Plante, A.F., Rasmussen, C., Schädel, C., Schimel, J. P., Sierra, C.A., Wagai, R. and Wieder, W.R.: Improving understanding of soil organic matter dynamics by triangulating theories, measurements, and models, Biogeochemistry, 140(1), 1–13, doi:10.1007/s10533-018-0478-2, 2018.

Braakhekke, M.C., Guggenberger, G., Schrumpf, M. and Reichstein, M.: Contribution of sorption, DOC transport and microbial interactions to the 14C age of a soil organic carbon profile: Insights from a calibrated process model, Soil Biology & Biochemistry, 88(C), 390–402, doi:10.1016/j.soilbio.2015.06.008, 2015.

Bradford, M.A., Wieder, W.R., Bonan, G.B., Fierer, N., Raymond, P.A. and Crowther, T.W.: Managing uncertainty in soil carbon feedbacks to climate change, Nature Climate Change, 6(8), 751–758, doi:10.1038/nclimate3071, 2016.

Broecker, W.S. and Olson, E.A.: Roadiocaronb from Nuclear Tests, II, Science, 132(3429), 712-721, doi:10.1126/science.132.3429.712. 1960.

Chen, J., Zhu, Q., Riley, W. J., He, Y., Randerson, J. T. and Trumbore, S. E.: Comparison With Global Soil Radiocarbon Observations Indicates Needed Carbon Cycle Improvements in the E3SM Land Model, J Geophys Res-Biogeo, 88(5), 390–17, doi:10.1029/2018JG004795, 2019.

Cherkinskii, A.E. Contemporary hypotheses of humification and radiocarbon analyses of some types of soils. Doklady Akademii nauk SSSR, 258(4), 993-996, 1981.

Crow, S.E., Swanston, C.W., Lajtha, K., Brooks, J.R. and Keirstead, H.: Density fractionation of forest soils: methodological questions and interpretation of incubation results and turnover time in an ecosystem context, Biogeochemistry, 85(1), 69–90, doi:10.1007/s10533-007-9100-8, 2007.

Desjardins, T., Andreux, F., Volkoff, B. and Cerri, C.C.: Organic carbon and 13C contents in soils and soil size-fractions, and their changes due to deforestation and pasture installation in eastern Amazonia, Geoderma, 61(1-2), 103–118, doi:10.1016/0016-7061(94)90013-2, 1994.

Goh, KM., Stout, JD., Rafter, TA. Radiocarbon enrichment of soil organic-matter fractions in New Zealand soils. Soil Science, 123(6), 385-391, 1977.

Golchin, A., Oades, J.M., Skjemstad, J.O., Clarke, P. Soil-Structure and Carbon Cycling. Australian Journal of Soil Research 32(5): 1043-1068. 1994.

Golchin, A., Oades, J.M., Skjemstad, J.O., Clarke, P. Structural and Dynamic Properties of Soil Organic-Matter as Reflected by C-13 Natural-Abundance, Pyrolysis MassSpectrometry and Solid-State C-13 Nmr-Spectroscopy in Density Fractions of an Oxisol under Forest and Pasture. Australian Journal of Soil Research 33(1): 59-76. 1995.

Harden, J.W., Hugelius, G., Blankinship, J.C., Bond-Lamberty, B., Lawrence, C.R., Loisel, J., Malhotra, A., Jackson, R.B., Ogle, S., Phillips, C., Ryals, R., Todd-Brown, K., Vargas, R., Vergara, S.E., Cotrufo, M.F., Keiluweit, M., Heckman, K.A., Crow, S.E., Silver, W.L., DeLonge, M. and Nave, L.E.: Networking our science to characterize the state, vulnerabilities, and management opportunities of soil organic matter, Global Change Biology, 348(2), 895–14, doi:10.1111/gcb.13896, 2017.

He, Y., Trumbore, S.E., Torn, M.S., Harden, J.W., Vaughn, L. J. S., Allison, S.D. and Randerson, J. T.: Radiocarbon constraints imply reduced carbon uptake by soils during the 21st century, Science, 355(6306), 1419–1424, doi:10.1126/science.aag0262, 2016.

Heckman, K., Lawrence, C.R. and Harden, J.W.: A sequential selective dissolution method to quantify storage and stability of organic carbon associated with Al and Fe hydroxide phases, Geoderma, 312, 24–35, doi:10.1016/j.geoderma.2017.09.043, 2018.

Heimann M. and Reichstein M. Terrestrial ecosystem carbon dynamics and climate feedbacks. Nature **451**, 289–292. 2018.

Jackson, R.B., Lajtha, K., Crow, S.E., Hugelius, G., Kramer, M.G. and Piñeiro, G.: The Ecology of Soil Carbon: Pools, Vulnerabilities, and Biotic and Abiotic Controls, Annual Review of Ecology, Evololution, & Systematics, 48(1), 419–445, doi:10.1146/annurev-ecolsys-112414-054234, 2017.

Jastrow, J.D., Amonette, J.E. and Bailey, V.L.: Mechanisms controlling soil carbon turnover and their potential application for enhancing carbon sequestration, Climatic Change, 80(1-2), 5–23, doi:10.1007/s10584-006-9178-3, 2006.

Kaiser, M., Ellerbrock, R.H., & Sommer, M. Separation of coarse organic particles from bulk surface soil samples by electrostatic attraction. Soil Science Society of America Journal, *73*(6), 2118-2130, 2009.

Kaiser, M., and Berhe, A.A.. "How does sonication affect the mineral and organic constituents of soil aggregates?—A review." Journal of Plant Nutrition and Soil Science 177, no. 4: 479-495, 2014

Kaiser, K. and Kalbitz, K.: Cycling downwards - dissolved organic matter in soils, Soil Biology & Biochemistry, 52(C), 29–32, doi:10.1016/j.soilbio.2012.04.002, 2012.

Keiluweit, M., Nico, P. S., Kleber, M. and Fendorf, S.: Are oxygen limitations under recognized regulators of organic carbon turnover in upland soils*?* Biogeochemistry, 127(2-3), 157–171, doi:10.1007/s10533-015-0180-6, 2016.

Khomo, L., Trumbore, S.E., Bern, C.R. and Chadwick, O.A.: Timescales of carbon turnover in soils with mixed crystalline mineralogies, SOIL, 3(1), 17–30, doi:10.5194/soil-3-17-2017, 2017.

Koven, C. D., Riley, W. J., Subin, Z. M., Tang, J. Y., Torn, M. S., Collins, W. D., Bonan, G. B., Lawrence, D. M. and Swenson, S. C.: The effect of vertically resolved soil biogeochemistry and alternate soil C and N models on C dynamics of CLM4, Biogeosciences, 10(11), 7109–7131, doi:10.5194/bg-10-7109-2013, 2013.

Lawrence, Corey R., Beem-Miller, Jeffery, Hoyt, Alison M., Monroe, Grey, Sierra, Carlos A., Stoner, Shane, Heckman, Katherine, Blankinship, Joseph C., Crow, Susan E., McNicol, Gavin, Trumbore, Susan, Levine, Paul A., Vindušková, Olga, Todd-Brown, Katherine, Rasmussen, Craig, Hick Pries, Caitlin E., Schädel, Christina, McFarlane, Karis, Doetterl, Sebastian, Hatté, Christine, He, Yujie, Treat, Claire, Harden, Jennifer W., Torn, Margaret S., Estop-Aragonés, Cristian, Berhe, Asmeret Asefaw, Keiluweit, Marco, Marin-Spiotta, Erika, Plante, Alain F., Thomson, Aaron, Schimel, Joshua P., Vaughn, Lydia J. S. and Wagai, Rota: International Soil Radiocarbon Database v1.0, Zendo, doi:10.5281/zenodo.2613911, 2019.

Le Quéré, C., Andrew, R.M., Friedlingstein, P., Sitch, S., Hauck, J., Pongratz, J., Pickers, P. A., Korsbakken, J.I., Peters, G.P., Canadell, J.G., Arneth, A., Arora, V. K., Barbero, L., Bastos, A., Bopp, L., Chevallier, F., Chini, L.P., Ciais, P., Doney, S.C., Gkritzalis, T., Goll, D. S., Harris, I., Haverd, V., Hoffman, F.M., Hoppema, M., Houghton, R.A., Hurtt, G., Ilyina, T., Jain, A.K., Johannessen, T., Jones, C.D., Kato, E., Keeling, R.F., Goldewijk, K.K., Landschützer, P., Lefèvre, N., Lienert, S., Liu, Z., Lombardozzi, D., Metzl, N., Munro, D.R., Nabel, J.E., Nakaoka, M.S., S.-I., Neill, C., Olsen, A., Ono, T., Patra, P., Peregon, A., Peters, W., Peylin, P., Pfeil, B., Pierrot, D., Poulter, B., Rehder, G., Resplandy, L., Robertson, E., Rocher, M., Rödenbeck, C., Schuster, U., Schwinger, J., Séférian, R., Skjelvan, I., Steinhoff, T., Sutton, A., Tans, P.P., Tian, H., Tilbrook, B., Tubiello, F.N., van der Laan-Luijkx, I.T., van der Werf, G.R., Viovy, N., Walker, A.P., Wiltshire, A.J., Wright, R., Zaehle, S. and Zheng, B.: Global Carbon Budget 2018, Earth Systems Science Data, 10(4), 2141–2194, doi:10.5194/essd-10-2141-2018, 2018.

Lehmann, J. and Kleber, M.: The contentious nature of soil organic matter, Nature, 1–9, doi:10.1038/nature16069, 2015.

Leinemann, T., Preusser, S., Mikutta, R., Kalbitz, K., Cerli, C., Höschen, C., Mueller, C.W., Kandeler, E., and Guggenberger G.: Multiple exchange processes on mineral surfaces control the

transport of dissolved organic matter through soil profiles, Soil Biology and Biochemistry, 118, 79-90, 2018.

Manzoni, S., Katul, G.G. & Porporato, A. Analysis of soil carbon transit times and age distributions using network theories. Journal of Geophysical Research 114, G04025, 2009.

Martel, YA., Paul, EA. Effects of cultivation on organic-matter of grassland soils as determined by fractionation and radiocarbon dating. Canadian Journal of Soil Science, 54(4), 419-426, 1974.

Masiello, C., Chadwick, O.A., Southon, J., Torn, M. and Harden, J.W.: Weathering controls on mechanisms of carbon storage in grassland soils. Global Biogeochem Cycles, 18(4), GB4023, doi:10.1029/2004GB002219, 2004.

Mathieu, J.A., Hatté, C., Balesdent, J. and Parent, É.: Deep soil carbon dynamics are driven more by soil type than by climate: a worldwide meta-analysis of radiocarbon profiles, Global Change Biology, 21(11), 4278–4292, doi:10.1111/gcb.13012, 2015.

Minasny, B., B.P. Malone, A.B. McBratney, D.A. Angers, D. Arrouays, A. Chambers, V. Chaplot, Z. Chen, K. Cheng, B.S. Das, D.J. Field, A. Gimona, C.B. Hedley, S. Young Hong, B. Mandal, B.P. Marchant, M. Martin, B.G. McConkey, V. L. Mulder, S. O'Rourke, A.C. Richer-de-Forges, I. Odeh, J. Padarian, K. Paustian, G. Pan, L. Poggio, I. Savin, V. Stolbovoy, U. Stockmann, Y. Sulaeman, C-C. Tsui, T-G. Vågen, B. van Wesemael, L. Winowiecki. Soil carbon 4 per mille. Geoderma, 292:59-86. 2017.

Moni, Christophe, D. Derrien, P-J. Hatton, Bernd Zeller, and M. Kleber. Density fractions versus size separates: does physical fractionation isolate functional soil compartments?. Biogeosciences 9, 12:5181-5197, 2012.

Oades, J. M. The Retention of Organic-Matter in Soils, Biogeochemistry, 5(1), 35–70, 1988.

O'Brien, B.J. and Stout, J.D. Movement and turnover of soil organic matter as indicated by carbon isotope measurements. Soil Biology and Biochemistry, 10:309–317. 1978.

Obrien, BJ. Soil organic-carbon fluxes and turnover rates estimated from radiocarbon enrichments. Soil Biology & Biochemistry, 16(2), 115-120, 1984.

Ohno, T., Heckman, K.A., Plante, A.F., Fernandez, I.J. and Parr, T.B.: [14]C mean residence time and its relationship with thermal stability and molecular composition of soil organic matter: A case study of deciduous and coniferous forest types. Geoderma, 308, 1–8, doi:10.1016/j.geoderma.2017.08.023, 2017.

Paul, E.A., Morris, S.J., and Böhm, S., The determination of soil C pool sizes and turnover rates: biophysical fractionation and tracers. In: Lal, R. (Ed.), Assessment Methods for Soil Carbon. Lewis Publishers, New York, pp. 193–206, 2001.

Plante, A. F., Conant, R. T., Stewart, C. E., Paustian, K. and Six, J.: Impact of soil texture on the distribution of soil organic matter in physical and chemical fractions, Soil Science Society of America Journal, 70(1), 287–296, 2006.

Poulton, P., J. Johnston, A. Macdonald, R. White, D. Powlson. Major limitations to achieving "4 per 1000" increases in soil organic carbon stock in temperate regions: Evidence from long-term experiments at Rothamsted Research, United Kingdom. Global Change Biology, 24:2563–2584. 2018.

Rumpel, C., and I. Kögel-Knabner, Deep soil organic matter—a key but poorly understood component of terrestrial C cycle, Plant Soil, *338*(1-2), 143–158, doi:10.1007/s11104-010-0391-5. 2010.

Sanderman, J., Baldock, J., Amundson, R. Dissolved organic carbon chemistry and dynamics in contrasting forest and grassland soils. Biogeochemistry 89(2): 181-198. 2008.

Scharpenseel, H.W. Radiocarbon dating of soils - problems, troubles, hopes. In: D.H. Yaalon (Ed) Paleopedology (pp. 77-96). ISSS and Israel Univ Press: Jerusalem. 1971

Schmidt, M.W.I., Torn, M.S., Abiven, S., Dittmar, T., Guggenberger, G., Janssens, I.A., Kleber, M., Kögel-Knabner, I., Manning, D.A.C., Nannipieri, P., Rasse, D.P., Weiner, S. and Trumbore, S.E.: Persistence of soil organic matter as an ecosystem property, Nature, 478(7367), 49–56, doi:10.1038/nature10386, 2011.

Schrumpf, M., Kaiser, K., Guggenberger, G., Persson, T., Kögel-Knabner, I., Schulze, E.-D. Storage and stability of organic carbon in soils as related to depth, occlusion within aggregates, and attachment to minerals. Biogeosciences 10, 1675-1691. 2013

Sierra, C., Müller, M. and Trumbore, S.E.: Models of soil organic matter decomposition: the SoilR package, version 1.0, 5(4), 1045–1060, doi:10.5194/gmd-5-1045-2012, 2012.

Sierra, C.A., Müller, M., and Trumbore, S.E.: Modeling radiocarbon dynamics in soils: SoilR version 1.1, Geosci. Model Dev., 7, 1919–1931, doi:10.5194/gmd-7-1919-2014, 2014.

Sierra, C.A., Müller, M., Metzler, H., Manzoni, S. and Trumbore, S.E.: The muddle of ages, turnover, transit, and residence times in the carbon cycle, Global Change Biology, 23(5), 1763–1773, doi:10.1111/gcb.13556, 2017.

Six, J. and Paustian, K.: Soil Biology & Biochemistry, Soil Biology & Biochemistry, 68(C), A4–A9, doi:10.1016/j.soilbio.2013.06.014, 2014.

Sollins, P., Kramer, M.G., Swanston, C., Lajtha, K., Filley, T., Aufdenkampe, A.K., Wagai, R. and Bowden, R. D.: Sequential density fractionation across soils of contrasting mineralogy: evidence for both microbial- and mineral-controlled soil organic matter stabilization, Biogeochemistry, 96(1-3), 209–231, doi:10.1007/s10533-009-9359-z, 2009.

Sollins, P., Swanston, C., Kleber, M., Filley, T., Kramer, M., Crow, S., Caldwell, B.A., Lajtha, K. and Bowden, R.: Organic C and N stabilization in a forest soil: Evidence from sequential density fractionation, Soil Biology & Biochemistry, 38(11), 3313–3324, doi:10.1016/j.soilbio.2006.04.014, 2006.

Sulman, B.N., Moore, J.A.M., Abramoff, R., Averill, C., Kivlin, S., Georgiou, K., Sridhar, B., Hartman, M.D., Wang, G., Wieder, W.R., Bradford, M.A., Mayes, M.A., Morrison, E., Riley, W. J., Salazar, A., Schimel, J.P., Tang, J. and Classen, A.T.: Multiple models and experiments underscore large uncertainty in soil carbon dynamics, Biogeochemistry, 141(2), 1–15, doi:10.1007/s10533-018-0509-z, 2018.

Swanston, C.W., M.S. Torn, P.J. Hanson, J.R. Southon, C.T. Garten, E.M. Hanlon, L. Ganio. Characterizing processes of soil carbon stabilization using forest stand-level radiocarbon enrichment. Geoderma 128:52–62, 2005.

Szymanski, L.M., G.R. Sanford, K. Heckman, R.D. Jackson and E. Marín-Spiotta. Conversion to bioenergy crops alters the amount and age of microbially-respired soil carbon. Soil Biology and Biochemistry 128: 35-44, 2019.

Tifafi, M., Camino-Serrano, M., Hatté, C., Morras, H., Moretti, L., Barbaro, S., Cornu, S. and Guenet, B.: The use of radiocarbon $^{14}$C to constrain carbon dynamics in the soil module of the land surface model ORCHIDEE (SVN r5165), Geosci. Model Dev., 11(12), 4711–4726, doi:10.5194/gmd-11-4711-2018, 2018.

Torn, M.S., Vitousek, P.M. and Trumbore, S.E.: The influence of nutrient availability on soil organic matter turnover estimated by incubations and radiocarbon modeling, Ecosystems, 8(4), 352–372, doi:10.1007/s10021-004-0259-5, 2005.

Trumbore, S.E. & Zheng, S. Comparison of Fractionation Methods for Soil Organic Matter 14C Analysis, Radiocarbon, 1996.

Trumbore, S.E.: Age of soil organic matter and soil respiration: Radiocarbon constraints on belowground C dynamics, Ecological Applications, 10(2), 399–411, 2000.

Trumbore, S.E.: Carbon respired by terrestrial ecosystems - recent progress and challenges, Global Change Biology, 12(2), 141–153, doi:10.1111/j.1365-2486.2005.01067.x, 2006.

Trumbore, S., M. Torn, and L. Smith. Constructing a database of terrestrial radiocarbon measurements, *Eos Trans. AGU*, 92(43), 376, doi:10.1029/2011EO430006. 2011.

Ziegler, S.E., Benner, R., Billings, S. A., Edwards, K.A., Philben, M., Zhu, X. and Laganière, J.: Climate Warming Can Accelerate Carbon Fluxes without Changing Soil Carbon Stocks, Frontiers in Earth Science, 5, 535–12, doi:10.3389/feart.2017.00002, 2017.

# Tables

**Table 1.** The number of data points currently included at each hierarchical level in ISRaD v1.0.

| Entries | Sites | Profiles | Layers | Fractions | Incubations | Interstitial | Fluxes |
|---------|-------|----------|--------|-----------|-------------|--------------|--------|
| 212 | 550 | 1854 | 7734 | 3910 | 1978 | 353 | 2119 |

**Table 2.** Details of some core calculations included with ISRaD_extra. Additional variables will be regularly added to ISRaD_extra, an up-to-date list can be found at soilradiocarbon.org.

| Operation | Purpose | Output |
|---|---|---|
| fill_dates | Radiocarbon calculations and unit conversions often require the year of measurement. | If no date is reported for fraction and/or incubation observation dates, this function replaces those empty cells with the mandatory layer observation date. |
| fill_14c | In some studies, only fraction modern (FM) units are reported | If no $\Delta^{14}C$ values are reported, they are calculated from FM and the measurement date. |
| fill_coords | Spatial coordinates are required to plot soil profiles and to extract geospatial data. Profile level coordinates are often not reported in publications. | If no spatial coordinates are specified for a profile, this function fills those cells with the site coordinates, which are required for template ingestion. |
| delta_delta | Delta-delta ($\Delta\Delta$ ) is the offset between the $\Delta^{14}C$ ratio of the atmosphere and that of a sample during the year of collection and is a useful way to compare radiocarbon data across a range of collection years. | This function calculates the $\Delta\Delta$ values for all radiocarbon measurements in the database, using the profile coordinates and the year of observation to extract an atmospheric radiocarbon value for the region of sample collection. The output is appended as a new variable, e.g. lyr_dd14c. |
| fill_FM | In some studies, only $\Delta^{14}C$ values are reported. | If no FM values reported, they are calculated from $\Delta^{14}C$ values. |
| CStocks | Measurements of carbon concentration are not, on their own, good estimates of the mass of carbon in soils. Bulk density and soil layer depth information is also needed. | If not measured directly, organic carbon concentration is filled with total carbon concentration (carbonates are accounted for only if reported). Then, with user-supplied bulk density, these values are used to calculate the mass of carbon in each soil layer (i.e., C stocks). |
| fill_expert | In some cases, data measured at one extent may be reasonably substituted at another extent for the purposes of conducting comparisons across incomplete datasets | Original reported data can be merged with expert suggested data to provide unreported bulk layer values. These estimates are not from original studies, and may be approximations, but they are useful for large-scale global analyses. |

| geospatial.climate | Across a wide range of datasets, basic climate variables are inconsistently measured and reported. The purpose of this function is to fill separate, geospatially-estimated, climate parameters with a consistent source and scale. | This function uses the site coordinates to pull climate, meteorological, soil and other parameters from known global scale source datasets. At present, we use climate data from WorldClim v1.4 (http://www.worldclim.org/bioclim) and soil classification and characteristics from ISRIC (https://www.isric.org/explore/soilgrids) |
| --- | --- | --- |

# Figure Captions

**Figure 1.** Conceptual diagram of an entry in the database. Each box represents a table in an entry; the horizontal bars distinguish the hierarchical levels of the database. Arrows show the hierarchical relationship between and among levels of the database. Time is considered at the profile level, as this is the coarsest spatial scale for which observational data are reported. Every time a profile is sampled a unique profile identifier must be generated, consisting of the profile name combined with the profile observation date, which is then linked to all measurements made at or below the profile level of the hierarchy.

**Figure 2.** An entity relationship diagram for the International Soil Radiocarbon Database (ISRaD). A short description of the required variables for each entity are shown along with the field name used in the database and the variable data type. Crow's foot connections with a straight line indicate mandatory daughter entities (one or more), whereas a crow's foot with an open circle indicates indicate optional (zero or more) daughter entities. The "*" indicates entries indicate keys, or linking variables, which are repeated at each successive level of the ISRaD hierarchy. The "^"indicates conditionally required values. A full list of non-required variables is available in the Template Information File.

**Figure 3.** One key feature of the ISRaD structure is the ability to classify and categorize data generated from diverse methods for fractionating soils. The ISRaD approach requires specification of the fractionation scheme applied, which may include but is not limited to: density (A), aggregate (B), and/or particle size (C) separations. In each of these examples, the fraction data is linked to a specific soil layer. Classification of the fractionation scheme along with several other fields that specify the nature of the fractionation method allow for an accurate partitioning of mass between the individual fractions, such that the total mass of the soil layer can be reconstructed. A proper accounting of mass attributable to each soil fraction, which in some cases may be derived from more complicated multistep or sequential fractionations (D), is essential in order to compare measurements across these diverse methods.

**Figure 4.** Geographic location of sites currently included in ISRaD v1.0. Circles that appear darker in color indicate multiple overlapping sites at the resolution of the map.

**Figure 5.** A simplified depiction of the ISRaD governance pyramid, where the scientific steering committee is responsible for approving major management decisions and data maintainers are responsible for implementing broad changes, but data contributors and users are the primary drivers of the evolution of the data product.

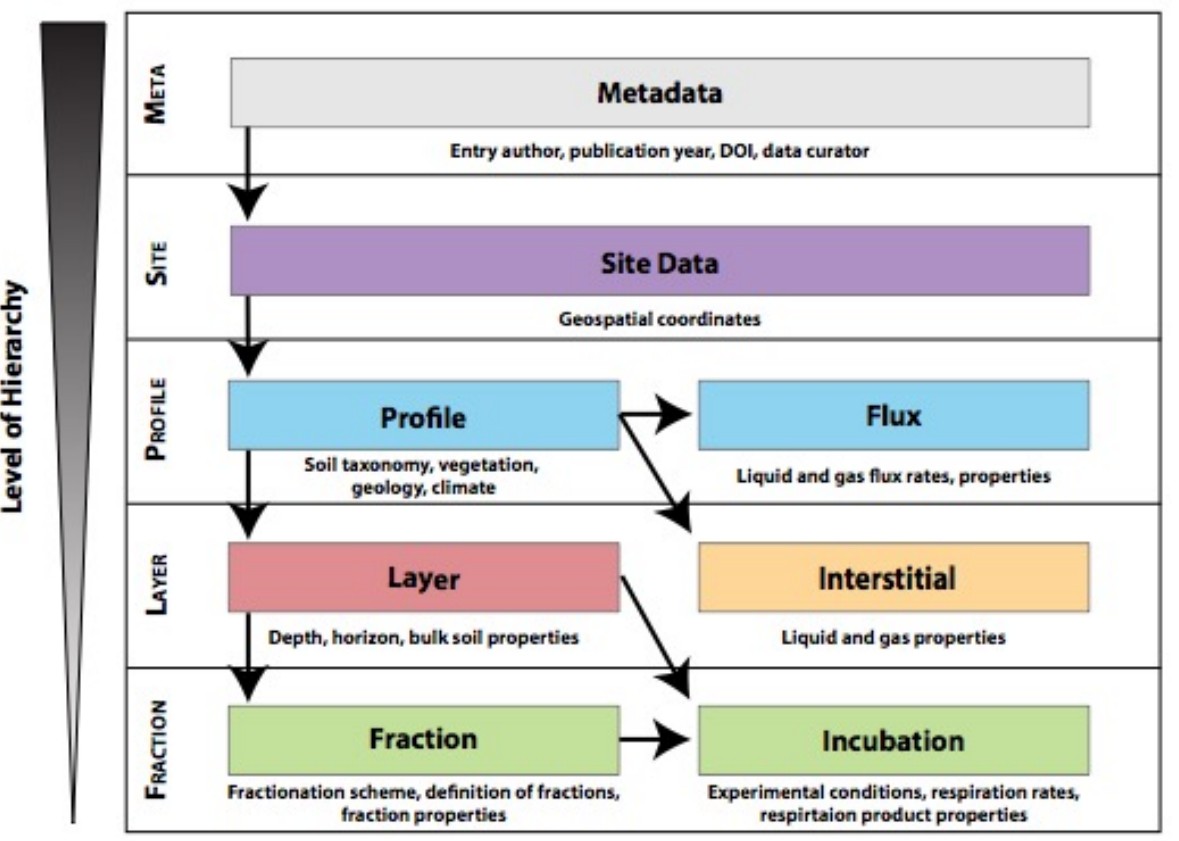

Figure 1.

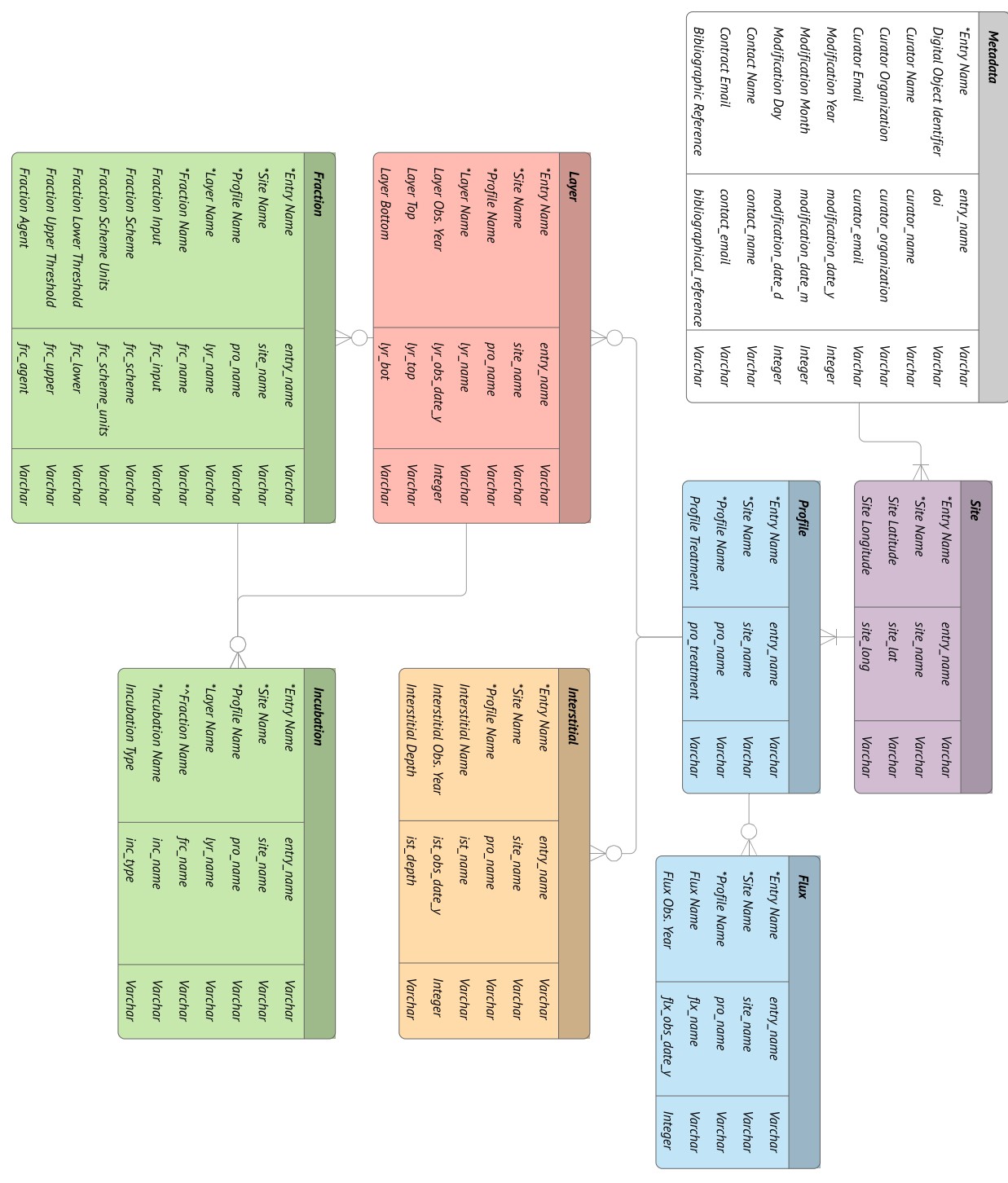

Figure 2.

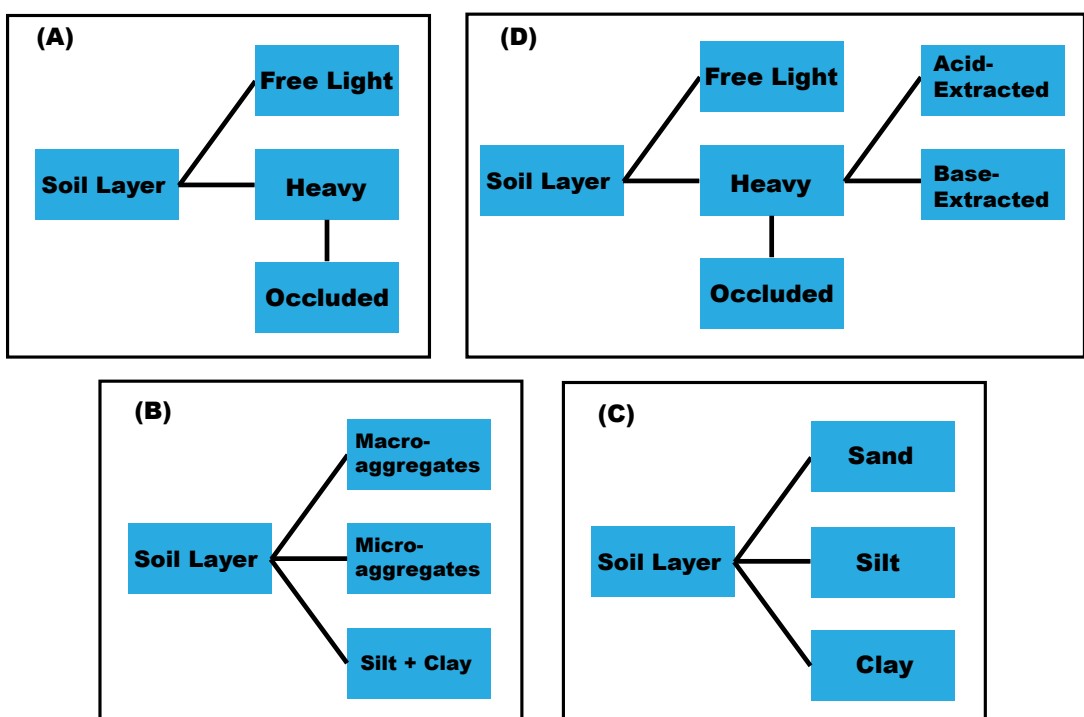

Figure 3.

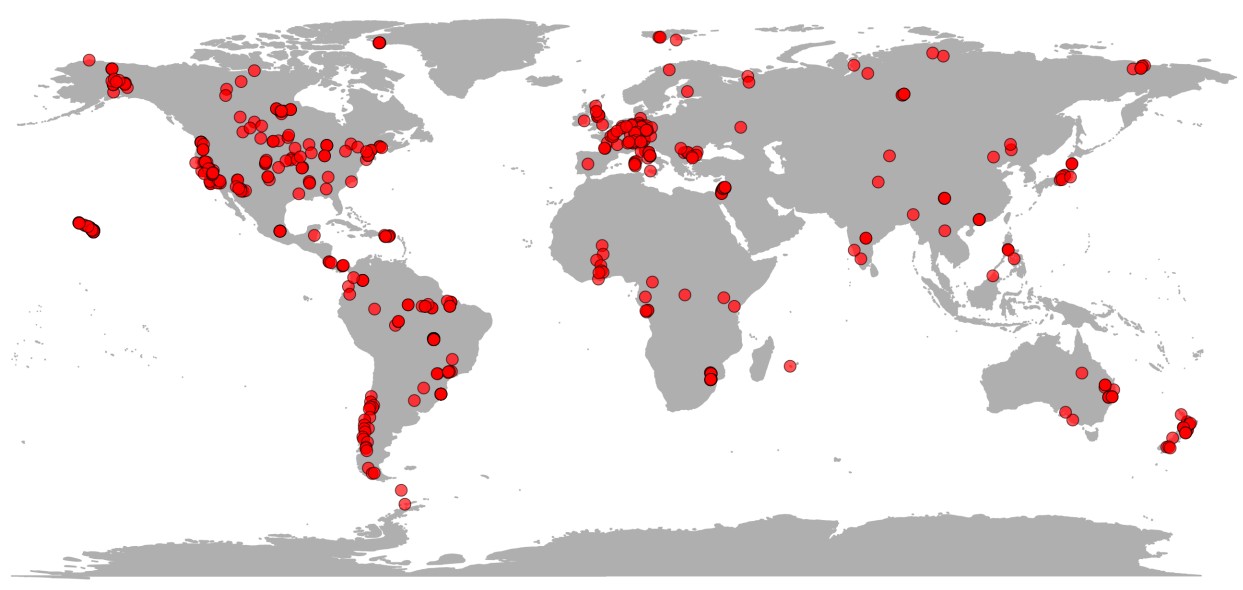

Figure 4.

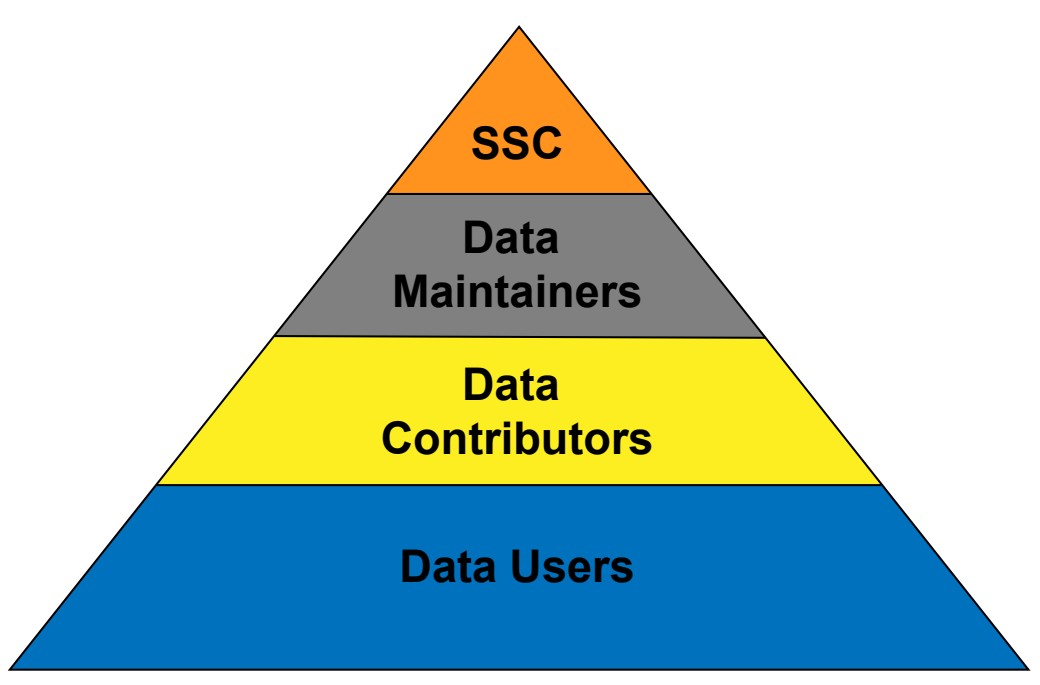

Figure 5.