# Peer review of "An open source database for the synthesis of soil radiocarbon data: ISRaD version 1.0"

_Earth System Science Data, 2019_

## Referee Comment (RC1) · Jonathan Sanderman (Referee) · 21 May 2019

This MS describes the International Soil Radiocarbon Database (ISRaD). ISRaD is really an active research community and should be lauded as a model for modern open science. The database itself is well described and easy to download and work with. The goal of ISRaD, and the focus of a large portion of this MS, is to make this database a living document that gains wide community acceptance and thus contributions.

Overall, the MS is very well written with only a few minor corrections listed below.

In terms of more major considerations, the only thing that I think is missing from this MS is a discussion about the different reporting conventions for 14C data and how the database deals with this. In particular, in order to compare 14C data across sites, some

[Figure]

consideration of sample collection date and sample analysis data might be required.

Minor corrections:

Abstract - Second to last sentence is an awkwardly phrased fragment. Please revise.

Line 27 – "these papers use and the uptake of bomb carbon" – I think "and" should be "of" here.

Figure 2 is mentioned in the text before Fig 1

———————————————————

---

## Referee Comment (RC2) · Anonymous Referee #2 · 25 May 2019

I consider the paper very interesting and the information presented very usefully. Some minor issues must be addressed before accepting this paper. Please, check my attached pdf and answer my questions. Some general questions: 1. How do you pretend to promote this paper? database? Courses? Meetings? 2. I consider that you should be really strict checking the homogenisation of the database. My big concern always is standardisation. I am very sad and disappointed when I use a database and for example, soil classification for each area is different, not updated or with not enough precision, I mean, order or sub-order. How do you pretend to correct that?
* * *

---

## Referee Comment (RC3) · Anonymous Referee #2 · 25 May 2019

[Figure]

[Figure]

[Figure]

[Figure]

Figure 1.

[Figure]

Figure 2.

[Figure]

[Figure]

Figure 3.

[Figure]

Figure 4.

[Figure]

[Figure]

Figure 5.

[referee-annotated manuscript omitted]

---

## Referee Comment (RC4) · Anonymous Referee #3 · 3 Jun 2019

This paper is very meaningful by developing the International soil radiocarbon database. The illustration is mostly clear. My first suggestion is that the authors may illustrate their data sources in a detail way and how you update your database. The other thing is that the authors need to illustrate how they synthesize data of different sources since the synthesis seems one of your point in your paper.The specific comments are as follows: Line 35, 'this database', do you mean the database your paper built Line 135: fig.2 should come after figure. 1 Line 145-146, a little confusing there is no legend of points in figure 4 Figure 5: it seems that this figure is too simple ?? Table 2, what do you mean by "Measurements of carbon concentration are not, on their own, good estimates of the mass of carbon in soils"

[Figure]

2019.

---

## Referee Comment (RC5) · Anonymous Referee #4 · 4 Jul 2019

The paper address an very important topic on soil carbon dynamics. The efforts as organized will be definitely contribute significantly to the community. I only had one minor question as follows: One of the purposes of ISRaD is a trial to gather soil radiocarbon data and trying to bring synthesis between different methods, to understand the turnover time, residence time, or mean age of carbon in soil. Although the paper has been introducing the ISRaD with a great amount of details, it seems to me the analysis of these data (e.g. bringing the synthesis) is not yet presented/discussed. It would be nice to see a brief demonstration on the spatially distributed points, showing the turnover time, residence time, mean age of carbon in soil, etc.

---

## Referee Comment (RC6) · Troy Baisden (Referee) · 10 Jul 2019

There is a great deal of enthusiasm in the growing soil radiocarbon community for a tool that makes existing data more accessible. This paper presents the latest and most significant effort to develop such a tool, ISRaD, authored by a group of leaders in the field. Overall, this paper is ideally suited to this journal. Its imperfections describe well the challenges faced in this field of research, and the reasons why the development of this database, observable at major conferences for a few years, has been such a substantial effort.

Below I document a number of areas of improvement for the manuscript and the database package. What I say less about is that the work-to-date and overall quality

of the manuscript here are very good and will generate great benefits through ongoing focus on the role of soil radiocarbon in managing a key part of the Earth's C cycle. The 8500 measurements included so far, valued at roughly $4.5 million USD, describe the scale of the problem, and the need for further progress in the development of a structured database to support this field of research. Downloading and browsing the database tables also emphasises this. However, there are significant opportunities to improve. These boil down to two things:

1. Because the ISRaD package was not on the CRAN repository for R packages as suggested in the manuscript during this review/discussion process, and the active Github site didn't seem to me to provide an easy/clear substitute, there is a little less checking and transparency than I would ideally like to see given the substantial effort that has gone into this work. This is most evident in that there is nothing resembling a set of worked examples that demonstrate use cases for the database.

2. The presentation appears to have an overemphasis on soil fractionation data and underemphasis on time series, other constraints enabled by the database, including particularly site level C flows such as NPP and soil respiration. This appears to be related to a view of a paradigm shift in the controls on soil organic matter dynamics that I find biased toward recent synthesis and in surprising ignorance of key original work decades earlier. A consequence of this is that the role of early radiocarbon work, that was often more thoughtful than recent studies (perhaps due to the higher relative cost of analysis) is underemphasised.

Overall the concerns related to (2) are significant, because unintentional biases in how scientists or teams of scientists were thinking when methods or datasets are created, selected or pruned can have long-lasting effects to obscure or wall off promising routes forward. Documentation via the scientific literature represents the last chance to correct or clarify any biases.

I provide additional detail and discussion in relation to particular lines in the manuscript.

L39-41 The sentence spanning these lines is problematic for several reasons. First, on the face of it, its assertion regarding bulk carbon appears to me to be disproven by Baisden et al (2001; 2013a, 2013b). This may simply be a matter of interpretation however – I also have trouble linking this sentence to what follows it, given what is possible in quantifying the stabilisation and turnover of carbon. Second, the confusion I see in this sentence may perhaps lie in what is meant by the word, "predict." To play devil's advocate on this point, I'll suggest that far more would be known and quantified if, since 2000, the field had followed the simple process of collecting and running time series bulk samples, and using math to separately measure the size and turnover rates of pools. In contrast, it doesn't seem that ongoing efforts to chemically and/or physically fractionate soil have led to clarity or application.

L42 "mean ages and cycling rates" are duplicative, since rate is the inverse of age given simple pools. Also, why imply "mean" ages? Mean ages, especially when used across distinct pools, or without time series data imply considerable risks of biased results (Baisden et al 2013a) so I would like to see see the community to be careful and precise in the use of this type of terminology.

L45 The introduction of transit time as a completely different measure is confusing. A great deal of work has included an understanding of transit, for example by explicitly attempting to model transport processes within soil. It may seem pedantic, but it quite important to understand that "transit" times are useful in systems where transport is important as a process. This distinction should either be left vague, noting that the measures differ somewhat, or be better expanded to recognise work focussed on transport. Useful examples include Elzein and Balesdent 1995, Baisden et al 2002; Baisden and Parfitt 2007, Sanderman et al 2008, and Jenkinson and Coleman 2008.

L48-49 This statement appears incorrect. It certainly has been shown mathematically tractable to separate distinct 'pools' without physically or chemically separating soil. For grassland soils, the comparisons in Baisden et al 2002, and further work in Baisden et al 2013 and 2013a make a fairly clear mathematical separation with time series samples is more efficient. Undoubtedly options may vary on this topic, but at a minimum the case for mathematical separation based on bulk samples has to be acknowledged as valid strategy. This is particularly true if total throughput of C through the ecosystems can be understood (Gaudinski et al 2000; Sierra et al 2012; Baisden and Keller 2013).

L51-52 The references given for the shifting view of controls on soil organic matter dynamics give an unfair impression of recent progress, using papers that do present useful recent synthesis. It seems remiss not to include earlier references, or at least Oades 1989. It would be preferable to include Golchin et al 1996 as well.

L57-59 It might be more accurate to say there are either one or three things here, but not two? If there is a 'fast' pool, and a slow 'pool' then different processes govern the turnover of each, so the two processes each need parameters. But it is equally important that the process of partitioning carbon flows into soil between the two pools be understood. Yet, I could also see another point of view, that there are typically more than two pools recognised in soil, so perhaps an understanding of partitioning only is intended here? Please clarify.

L 68 It seems slightly odd not to have pioneering or earlier exemplars of density fraction in this list. Various students of Oades, and particular series of papers published in 1995-7 by Golchin. Keep in mind that many of these methods were not developed specifically for radiocarbon.

L84 It is interesting here to see version 1.0 (Sierra et al 2012) rather than version 1.1 (Sierra et al 2014) of SoilR referenced. Please see the note below regarding L102 about an interface to soilR. If the goal of the database is to allow improved testing of hypotheses representing understanding of soil carbon dynamics, it seems SoilR should provide an ideal mechanism for implementation. It would be good to see more clarity of thought on achieving this, including a reference to the later version of SoilR.

L88-89 It may be worthwhile considering earlier references to DOM such as Sanderman et al 2008. I say this in part, because what is said in this paper may guide the

use of the database, and it would be worrisome to neglect early studies containing compelling radiocarbon results.

L102 Here again, I'd propose there is a collective forgetfulness of what was well established in the literature by the 1990s in terms of paradigms of soil organic matter dynamics. These have been reinforced by review and synthesis in recent decades, but this is not a reason to neglect early radiocarbon work that had already largely incorporated the paradigms promoted in this introduction. Therefore, it is odd to see early work that established overall constraint of carbon dynamics in well-studied systems neglected here. Such work can provide useful examples of how to construct strategies for constraining carbon dynamics. The obvious examples driven entirely by radiocarbon are Gaudinski et al 2000 (in relation to followup by Sierra et al 2012) and the set of work in Baisden et al 2002, 2002a, 2003. A second issue the lack of reference to or inclusion of literature using tracer carbon, or natural abundance stable C isotope ratios. Finally, a weakness in papers on recent paradigms is the importance of closing the partitioning and turnover of soil carbon by constraining the overall flow of C through the system via NPP or respiration. This is a strength of SoilR (Sierra et al 2014) so, as noted above, I would like to encourage the authors to consider what link might be made between these two R packages. This is covered to some degree in L225-231 but not with explanation of the value of or rationale for such constraints.

L131 It would be good to clarify here that the site accessible via the soilradiocarbon.org address does not appear to have an R-shiny interface or some other "web interface" to the data running. Either the words "web interface" should be changed to "web site", or an exact address to a "web interface" should be provided.

L270-279 It is good to see these items related to density fractionation included specifically. However, does it make sense to include/explain these stored values but not include the degree to which the sonication method has enabled isolation of occluded vs free light fraction, again originally detailed by the Golchin work to adapt density fractionation to the paradigms the authors promoted at the beginning of this manuscript's

introduction?

L320 The web interface again appears to be a regular website rather than a web database interface?

L322 This web address only goes through with http:// and not with https://

L332 The ISRaD package was not available at cran.org as implied in this text. The Github version indicates changes. Although these changes are probably minor I was disappointed to find that there was not a version tagged to support this review process.

L501 Here again the "web interface" is mentioned. It seems worth noting here that this link appears to lead to a fairly standard website with a static file download for a database table, rather than an interface to the database.

What's missing? There seems to be neither an accounting of the spread of categories or types the data already in the database represent, or what weaknesses (gaps) can be described. Similarly, there is a rather technical description of data entry, but not a description of how substantial historic datasets might be brought into the database. An additional but admittedly problematic question is whether the extent of available published data not in the database can be better quantified and described. I encourage some discussion of these opportunities for improvement.

References cited (where not in discussion paper)

Baisden WT, Parfitt RL, Ross C, Schipper LA, Canessa S 2013. Evaluating 50 years of time-series soil radiocarbon data: towards routine calculation of robust C residence times. Biogeochemistry 112(1-3): 129-137.

Jenkinson DS, Coleman K 2008. The turnover of organic carbon in subsoils. Part 2 Modelling carbon turnover. European Journal of Soil Science 59: 400-413.

Baisden WT, Canessa S 2013a. Using 50 Years of Soil Radiocarbon Data to Identify Optimal Approaches for Estimating Soil Carbon Residence Times. Nuclear Instruments and Methods in Physics Research Section B: Beam Interactions with Materials and Atoms – AMS 12 Proceedings 294: 588-592.

Baisden WT, Keller ED 2013. Synthetic Constraint of Soil C dynamics Using 50 Years of Radiocarbon and Net Primary Production (NPP) in a New Zealand Grassland Site. Radiocarbon 55(2-3): 1071-1076.

Baisden WT, Parfitt RL 2007. Bomb C-14 enrichment indicates decadal C pool in deep soil? Biogeochemistry 85(1): 59-68.

Baisden WT, Amundson R, Brenner DL, Cook AC, Kendall C, Harden J 2002. A Multi-Isotope C and N Modeling Analysis of Soil Organic Matter Turnover and Transport as a Function of Soil Depth in a California Annual Grassland Soil Chronosequence. Global Biogeochemical Cycles 16((4)): 1135, doi: 1029/2001GB001823.

Baisden WT, Amundson R, Cook AC, Brenner DL 2002a. The turnover and storage of C and N in five density fractions from California annual grassland surface soil. Global Biogeochemical Cycles 16: doi: 1029/2001GB001822.

Baisden WT, Amundson R 2003. An analytical approach to ecosystem biogeochemistry modeling. Ecological Applications 13(3): 649-663.

Elzein A, Balesdent J 1995. Mechanistic simulation of vertical distribution of carbon concentrations and residence times in soils. Soil Science Society of America Journal 59: 1328–1335.

Sanderman J, Baldock J, Amundson R 2008. Dissolved organic carbon chemistry and dynamics in contrasting forest and grassland soils. Biogeochemistry 89(2): 181-198.

Gaudinski JB, Trumbore SE, Davidson EA, Zheng SH 2000. Soil carbon cycling in a temperate forest: radiocarbon-based estimates of residence times, sequestration rates and partitioning of fluxes. Biogeochemistry 51(1): 33-69.

Golchin A, Clarke P, Baldock JA, Higashi T, Skjemstad JO, Oades JM 1997. The ef-
fects of vegetation and burning on the chemical composition of soil organic matter in a volcanic ash soil as shown by C-13 NMR spectroscopy .1. Whole soil and humic acid fraction. Geoderma 76(3-4): 155-174.

Golchin A, Clarke P, Oades JM 1996. The heterogeneous nature of microbial products as shown by solid-state C-13 CP/MAS NMR spectroscopy. Biogeochemistry 34(2): 71-97.

Golchin A, Clarke P, Oades JM, Skjemstad JO 1995. The Effects of Cultivation on the Composition of Organic-Matter and Structural Stability of Soils. Australian Journal of Soil Research 33(6): 975-993.

Golchin A, Oades JM, Skjemstad JO, Clarke P 1994. Soil-Structure and Carbon Cycling. Australian Journal of Soil Research 32(5): 1043-1068.

Golchin A, Oades JM, Skjemstad JO, Clarke P 1994. Study of Free and Occluded Particulate Organic-Matter in Soils by Solid-State C-13 Cp/Mas Nmr-Spectroscopy and Scanning Electron-Microscopy. Australian Journal of Soil Research 32(2): 285-309.

Golchin A, Oades JM, Skjemstad JO, Clarke P 1995. Structural and Dynamic Properties of Soil Organic-Matter as Reflected by C-13 Natural-Abundance, Pyrolysis Mass-Spectrometry and Solid-State C-13 Nmr-Spectroscopy in Density Fractions of an Oxisol under Forest and Pasture. Australian Journal of Soil Research 33(1): 59-76.

Oades JM 1989. An introduction to organic matter in mineral soils. In: Dixon JB, Weed SB ed. Minerals in Soil Environments. Madison, WI, Soil Science Society of America. Pp. 89-159.

Sierra CA, Trumbore SE, Davidson EA, Frey SD, Savage KE, Hopkins FM 2012. Predicting decadal trends and transient responses of radiocarbon storage and fluxes in a temperate forest soil. Biogeosciences 9(8): 3013-3028.

Sierra CA, Müller M, Trumbore SE 2014. Modeling radiocarbon dynamics in soils: SoilR version 1.1. Geosci. Model Dev. 7(5): 1919-1931.

---

## Author Comment (AC1) · 16 Aug 2019

Reviewer #1 (Jonathan Sanderman) Comment: This MS describes the International Soil Radiocarbon Database (ISRaD). ISRaD is really an active research community and should be lauded as a model for modern open science. The database itself is well described and easy to download and work with. The goal of ISRaD, and the focus of a large portion of this MS, is to make this database a living document that gains wide community acceptance and thus contributions. Overall, the MS is very well written with only a few minor corrections listed below. In terms of more major considerations, the only thing that I think is missing from this MS is a discussion about the different reporting conventions for 14C data and how the database deals with this. In particular, in order to compare 14C data across sites, some consideration of sample collection

date and sample analysis data might be required.

Response: We thank Dr. Sanderman for his thoughtful and constructive comments and we strongly agree with his assessment that a primary goal is to make this a living document to facilitate community participation.

With regard to 14C reporting conventions, we dictate that radiocarbon measurements are reported in standard units of fraction modern and/or Delta 14C. Further, we require that measurement date (year of measure at minimum) is also reported. These requirements ensure that all radiocarbon data in the database can be converted between the two standard units of measure. If the data is reported as a calibrated date, it cannot be included in ISRaD. Uncalibrated radiocarbon ages can be converted to fraction modern values in order to meet our submission requirements. For studies that use alternate units, we request that the person ingesting the data convert the measurements reported to the acceptable units, and provide detailed resources to do so on our website FAQ (https://international-soil-radiocarbon-database.github.io/ISRaD/template_faq/), accessed following the options Contribute»Radiocarbon Data. As requested, we would add a short section (inserted as section 2.3 – see below) to the text providing guidance on the reporting conventions accepted.

To make the data easily accessible to users, we provide built-in unit conversions between the two accepted standard radiocarbon unit systems (fraction modern and Delta 14C). When the data is ingested, we request contributors to ISRaD only fill in the units originally reported, along with the year that the radiocarbon data were measured. In some cases, in order to facilitate these conversions for older studies that report only Delta 14C, we must estimate the date that the samples were measured. These automatic unit conversions are then performed separately and included in the "ISRaD_extra" data object, which is part of the ISRaD R-package and available for direct download from the website.
New Text for Manuscript: "2.3 Radiocarbon Data – Reporting Conventions Radiocarbon measurements of environmental samples have a long history, much of which is reviewed in Trumbore (2009) including common units. Radiocarbon data ingested to ISRaD are required to adhere to some basic reporting conventions. First, measurements of radiocarbon may be reported in units of either fraction modern (FM) or Delta 14C. Other units are not supported at this time – for example, calibrated radiocarbon dates are not accepted, as the calibration curves are evolving over time. Such calibrated ages make sense only for certain fractions (e.g. macrofossils found in soils), and do not make sense in the context of most soil organic matter, which is an open system for carbon. For datasets where radiocarbon is reported in units other that FM or Delta 14C, it is up to the data curator or original author of the dataset to convert the reported values to one or both of the permitted units. Second, the year of measurement for each radiocarbon value must also be reported so that values may be internally converted between the two accepted units. In addition to these basic requirements, there are several other optional fields pertaining to radiocarbon data. These include the radiocarbon laboratory (rc_lab); the laboratory number (rc_lab_number), a unique identifier issued by each AMS facility; the analytical error reported for each measurement reported by most laboratories; and the environmental standard deviation of replicate samples (if analyzed). These variables are not required for data submission but should be included if they are available."

Minor corrections: Abstract - Second to last sentence is an awkwardly phrased fragment. Please revise. Response: We will revise this sentence.

Line 27 – "these papers use and the uptake of bomb carbon" – I think "and" should be "of" here. Response: We will change as requested

Figure 2 is mentioned in the text before Fig 1 Response: We will correct the order of figure references in the text

Reviewer #2 I consider the paper very interesting and the information presented very

usefully. Some minor issues must be addressed before accepting this paper. Please, check my attached pdf and answer my questions. Some general questions:

1. How do you pretend to promote this paper? database? Courses? Meetings?

Response: We thank the reviewer for their thoughtful comments and questions. We have already begun promoting the database and the associated paper through organized sessions and talks at international meetings. To date, we have presented (or are slated to present) aspects of this work at the International Radiocarbon Conference, the American Geophysical Union, the European Geophysical Union, the Soil Science Society of America, the Goldschmidt meeting of the Geochemical Society, etc. In addition, we have co-organized workshops, seminars, and "hack-a-thons" (often in collaboration with the International Soil Carbon Network) to provide more detailed guidance on the use of this resource. In the future, we plan to organize short-courses to provide even more detailed training and demonstration for stake-holders. Relatedly, we are constantly improving the documentation and training tools provided through our website and we actively manage the GitHub issues page, which serves to provide community based technical support feedback on problems or questions related to ISRaD (as described in section 4.3).

2. I consider that you should be really strict checking the homogenisation of the database. My big concern always is standardisation. I am very sad and disappointed when I use a database and for example, soil classification for each area is different, not updated or with not enough precision, I mean, order or sub-order. How do you pretend to correct that?

Response: This is a very important question/concern and is something that we have thought a lot about. There are at least two contrasting approaches for maintaining databases such as ISRaD. One approach is to strictly curate and control all data that is entered into the database. This approach results in standardized data but is resource intensive, requiring a large degree of management and quality control, and can limit

the ingestion of data that does not meet the standards. An opposite approach is to ingest any and all data but to give little attention to standardization. This allows for a more complete synthesis of available datasets but can lead to confusion for data users and erroneous comparisons, if data types are not clearly documented.

Our approach has been to take the middle road. We have constructed rules for ingestion that force a minimum degree of standardization (e.g. through pick-lists and required units for certain variables) and we have built tools to automate aspects of the quality control process – though we still utilize human reviewers for each dataset that is ingested. Specifically, our data review process starts with the requirement that submitted templates pass the automated QA/QC test that ensures linking variables connecting observations from each data table (i.e., hierarchical level) match, required fields have been entered, and values fall with in the acceptable ranges. Once the template has passed this automated test, the data contributor submits the template and data source to ISRaD and it is then passed to an expert reviewer, who checks several aspects of each data table to ensure ISRaD standard practices for categorizing data are met.

For some variables we allow a range of input types – requiring additional fields to document the type reported in each entry. We feel that our approach allows for an acceptable degree of standardization but limits the number of preexisting datasets that are rejected. We feel that by capturing and documenting as many of the available datasets as is reasonable, we will retain the ability to convert diverse observations to a standardized unit or class in the future. Importantly, the methodologies that might be used for such conversions can be clearly documented and the raw information is retained in the event that such conversions prove to be undesirable in the future.

Soil classification is a very good example and is a parameter that we have had long discussions about how to handle. We agree that forcing a standardized system would result in a dataset that is easier to compare. However, many of the datasets that we have compiled use different classification systems and converting from one to the other

is not always straightforward. Our approach has been to record the raw information reported in each dataset (along with the classification system used). This provides us the opportunity to create a scripted tool for converting all (or most observations) to a single classification system and minimizes the potential for human errors (that cannot be undone) in the process.

Line Item Comments

Abstract Line 6. Missing comma; replace "describe" with "present" Response: This will be corrected as requested

Line 10. Missing comma Response: This will be corrected as requested

Last Line. Do not add references in the abstract. Response: A reference to the dataset in the abstract is a requirement of the EESSD.

Main Text. Line 3. Missing references. Response: We will add references as requested

Line 11. Specify how deep soils contribute to carbon storage Response: We will add additional detail as follows (proposed changes to manuscript text shown in blue): "Additionally, many studies and models focus on only the top 0.5 m of soil or less, despite deeper soils contributing a significant proportion of SOM storage by way of low carbon concentrations but large deep soil mass (Rumpel and Kögel-Knabner, 2010)."

Line 15. Add something related to sustainable development goals, land management plans, or policy. Response: We will add additional text referencing these important issues as follows: "There is an urgent need to synthesize a wide variety of soils data to model the role of soil in the climate system (Bradford et al., 2016), to develop more data-driven estimates of soil health (Harden et al., 2017), to inform policy and land management plans that preserve and enhance soil carbon storage (Minasny et al. 2017; Poulton et al. 2018), and to extend our detailed understanding of soil developed from observations made at the profile scale to both regional and global extents."

Line 24. Refs. Response: Additional references will be added as requested

Line 27. Replace "papers" with "manuscripts" or "researchers" Response: We will replace "papers" with "studies."

Line 29. Define accelerator mass spectrometry. Response: Additional detail will be added to this statement as follows: "The advent of accelerator mass spectrometry, a method that measures 14C atoms in a sample by accelerating them to high energy, in the 1980's allowed for radiocarbon analysis using milligrams of carbon instead of grams, while simultaneously increasing sample throughput (Trumbore, 2009)."

Line 30. References. Response: Additional references will be added

Line 34. Add approximation of paper numbers. Response: We will add an approximation as follows:"These applications have led to an explosion in the number of publications with radiocarbon measurements from soil, increasing from a few dozen papers annually during the 1980's to more than 150 per year in the last decade (based on a search for papers with "soil" and "radiocarbon" as keywords in Google Scholar)."

Line 62. Refs. Response: This sentence will be modified by request of other reviewers and an additional reference will be added.

Line 67. Refs. Response: We will add references as requested.

Line 71. Why "? Response: We will add detail to clarify this sentence as follows: "The "light" fraction of soil material that floats in a dense solution (e.g., sodium polytungstate) or gets picked up by electrostatic attraction (Kaiser et al., 2009) is sometimes used as a proxy for rapidly-cycling SOM, as this material is generally observed to have a shorter mean residence time compared with the bulk soil average, while the "heavy" or dense material is used as a proxy for mineral-associated SOM, which is assumed to cycle more slowly (e.g., Sollins et al., 2009)."

Lines 109 to 113. Please split up. Response: As requested, we will split up this sentence.

Line 118. Missing comma Response: We will insert the missing comma as suggested

Line 161. I would mention this figure before or add a flow chart before. Better to follow the text Response: We will change the order of figure referencing in the text to correct this problem

Line 199. Missing space. Response: Space inserted

Line 235. You should insist investigators to use a specific soil classification, updated and leave clear which order is described.

Response: This is a good suggestion and something our leadership team has discussed extensively. We have, however, decided that requiring a specific soil classification is not appropriate for an international database. We cannot reasonably force scientists to learn and apply a new classification system, which they may not otherwise utilize. Instead, we are developing tools to convert user input from one classification system to another. This approach will be scripted and thus, there will be a traceable record of how the conversions are applied. This will ensure that all conversions follow the same rule-set and will allow for updates/modifications as needed. One of the reasons, we chose to take this approach is that for many datasets soil classifications are either not reported or are reported in an alternate classification system. If we require all datasets to utilize single classification system, we are dictating that the person entering the data (who may not be the original author of the study) must make the conversion. We feel it is more objective to record the original data and apply soil classification conversions uniformly across all datasets.

Line 369. Is it updated regularly with new papers? Response: This is a good question. At present there is no regular schedule over which this Soil Carbon Information Hub is updated. That said, we do anticipate providing updates periodically. We will formalize a regular update schedule and include that detail in the manuscript and on the SCIH.

Reviewer #3 This paper is very meaningful by developing the International soil radiocarbon database. The illustration is mostly clear. My first suggestion is that the authors may illustrate their data sources in a detail way and how you update your database.
The other thing is that the authors need to illustrate how they synthesize data of different sources since the synthesis seems one of your point in your paper. The specific comments are as follows:

Response: We thank the reviewer for their valuable input on this manuscript. The data sources currently included in the current version of the database can be found here: https://github.com/International-Soil-Radiocarbon-Database/ISRaD/blob/master/ISRaD_data_files/database/credits.md We will add additional text to the manuscript to make clear that this feature is available and always up to date. Our process for the synthesis of different data sources is based on entry of data into the ISRaD data template. Specific details on how contribute data can be found at our website here: https://international-soil-radiocarbon-database.github.io/ISRaD/contribute/. We feel that many of these specifics are beyond the scope of the manuscript, though the general process is described in section 2.3 of the manuscript.

Line 35, 'this database', do you mean the database your paper built. Response: Yes, we will add additional language to clarify that we mean the ISRaD database.

Line 135: fig.2 should come after figure. 1. Response: This point was also made by another reviewer; we will switch the order of the figures in the text.

Line 145-146, a little confusing there is no legend of points in figure 4. Response: We will add language to this section to clarify that the flagging of data from prior synthesis studies happens within the database and is not necessarily represented in the figures for this paper.

Figure 5: it seems that this figure is too simple ?? Response: We agree that this is a very simple figure. However, it is an accurate representation of the ISRaD user structure that is also quite simple.

Table 2, what do you mean by "Measurements of carbon concentration are not, on their

own, good estimates of the mass of carbon in soils" Response: We will add language in the table to clarify our point, which is that measurements of carbon concentration must be combined with measurements of soil bulk density and layer thickness in order to adequately estimate the total carbon mass contained within soil layers.

Reviewer #4 The paper address an very important topic on soil carbon dynamics. The efforts as organized will be definitely contribute significantly to the community. I only had one minor question as follows: One of the purposes of ISRaD is a trial to gather soil radiocarbon data and trying to bring synthesis between different methods, to understand the turnover time, residence time, or mean age of carbon in soil. Although the paper has been introducing the ISRaD with a great amount of details, it seems to me the analysis of these data (e.g. bringing the synthesis) is not yet presented/discussed. It would be nice to see a brief demonstration on the spatially distributed points, showing the turnover time, residence time, mean age of carbon in soil, etc.

Response: We thank the reviewer for their suggestion and agree that an analysis of the data is warranted and of interest to the soil science community. However, that work goes beyond the intended scope of this manuscript, which is to simply present the dataset and describe the infrastructure available to interface with the data. Furthermore, the variables mentioned by the reviewer – turnover time, residence time, and mean age – are derived from the raw data we have compiled and are subject to specific assumptions and methodologies used for these calculations. In other words, while the goal of the database is to facilitate such comparisons, we do not provide such calculations as part of the dataset. Such comparisons should be conducted by individual researchers or research groups and require a complete documentation of the methodology used. We hope that as these efforts are undertaken, the results can be added as a new data table to the database, but the more immediate goal of this work is to synthesize and document the raw data that are available.

Reviewer #5 (Troy Baisden) There is a great deal of enthusiasm in the growing soil radiocarbon community for a tool that makes existing data more accessible. This paper

presents the latest and most significant effort to develop such a tool, ISRaD, authored by a group of leaders in the field. Overall, this paper is ideally suited to this journal. Its imperfections describe well the challenges faced in this field of research, and the reasons why the development of this database, observable at major conferences for a few years, has been such a substantial effort. Below I document a number of areas of improvement for the manuscript and the database package. What I say less about is that the work-to-date and overall quality of the manuscript here are very good and will generate great benefits through ongoing focus on the role of soil radiocarbon in managing a key part of the Earth's C cycle. The 8500 measurements included so far, valued at roughly $4.5 million USD, describe the scale of the problem, and the need for further progress in the development of a structured database to support this field of research. Downloading and browsing the database tables also emphasises this. However, there are significant opportunities to improve. These boil down to two things:

Response: We are very appreciative of the time and effort that Dr. Baisden has contributed to providing constructive input on ways to improve ISRaD. Below, we address each of his suggestions in detail.

1. Because the ISRaD package was not on the CRAN repository for R packages as suggested in the manuscript during this review/discussion process, and the active Github site didn't seem to me to provide an easy/clear substitute, there is a little less checking and transparency than I would ideally like to see given the substantial effort that has gone into this work. This is most evident in that there is nothing resembling a set of worked examples that demonstrate use cases for the database.

Response: We apologize for the difficulties accessing the ISRaD package through the CRAN repository. CRAN's technical requirements have resulted in a time-consuming delay in updating the package in the repository any time that we added new data to the database. We are currently addressing this issue and hope to have it resolved shortly. One major change we are making is to change the way the data is served when using the R-package. Instead of including the database as a part of the package, we now

include tools required to upload the latest version of the database in the R-package. This approach reduces the size of the R-package and allows users to more easily update the version of the database that they are using within R without reinstalling the R-package.

The GitHub repository does provide nearly full transparency for how the database is structured or implemented. That said, we agree with Dr. Baisden that it is not the most user-friendly platform for doing so, which is why most of our tutorials leverage the R-based data access and manipulation. With that in mind we have several basic "worked examples" demonstrating the use of the database, which are available through the website. For example, based on the reviewer's suggestion, we have now included a vignette on downloading and accessing the development version of the R package, and generating the plots used in this manuscript (https://international-soil-radiocarbon-database.github.io/ISRaD/rpackage/).

2. The presentation appears to have an overemphasis on soil fractionation data and underemphasis on time series, other constraints enabled by the database, including particularly site level C flows such as NPP and soil respiration. This appears to be related to a view of a paradigm shift in the controls on soil organic matter dynamics that I find biased toward recent synthesis and in surprising ignorance of key original work decades earlier. A consequence of this is that the role of early radiocarbon work, that was often more thoughtful than recent studies (perhaps due to the higher relative cost of analysis) is underemphasised.

Response: We appreciate Dr. Baisden's concern regarding an apparent bias of the database toward soil fraction data, seemingly at the expense of the other constraints he has mentioned. However, it is our opinion that this perceived bias is a product of the database infrastructure required to accommodate soil fractionation data and not an oversight, as suggested. In fact, the database has been designed to accommodate measures of all of the variables that he has listed. For example, soil respiration measurements and associated radiocarbon values are incorporated in the Flux Data Table

(i.e., described in section 2.2.4) and DOC fluxes can be reported in the Interstitial Data Table (i.e., section 2.2.6). Similarly, time series observations can be (and are) reported for any of these measurements (including bulk solids or fractions) by incorporation of observations categorized identically, except for the date or time of observation.

Although the database can accommodate these other observation types, there are at least two related factors that drive this perception of bias toward soil fraction data. First, the early versions of the database were facilitated by a USGS Powell Center synthesis effort, which specifically targeted soil fraction data. As such, the emphasis was initially geared towards including studies that had fractionation data with associated radiocarbon measurements. Since that early effort we have greatly expanded the scale and scope of the database so as to allow for recording of all soil radiocarbon observations and other groups are currently expanding, for example to allow synthesis efforts on incubation data. Second, the infrastructure required to accommodate, and record soil fractionation methods is inherently more complex, as a result of the diversity of methodologies used, and therefore requires more details to describe.

Overall the concerns related to (2) are significant, because unintentional biases in how scientists or teams of scientists were thinking when methods or datasets are created, selected or pruned can have long-lasting effects to obscure or wall off promising routes forward. Documentation via the scientific literature represents the last chance to correct or clarify any biases.

Response: We do not claim to have been exhaustive in our synthesis efforts and again our efforts to compile datasets have been shaped by our own research interests as well as the interests of researchers whose work proceeded our own (i.e., Matheiu et al., 2015 and He et al., 2017). This is in part also driven by the funding sources that allowed us to join forces and make joint progress on a repository that we hope will grow and evolve with time as new researchers with new interests add to the data in it. We have constructed and described a database platform that can accommodate all types of data and we have put forth a good faith effort to include as many of the

available datasets as can be found in the published literature. However, this product is intended to be a living community resource that continues to grow in ways dictated by those who chose to participate. In this regard, this manuscript is intended to advertise the existing dataset but, perhaps more importantly, to attract new participants who can add additional data focused around their particular area of interest. We believe the necessary infrastructure is in place to accommodate underrepresented data types or adapt to new ones. With regard to the under-emphasis of early datasets, we would also note that because our requirements for data ingestion (i.e., section 2.2 and Figure 2), the ingestion of some older datasets is delayed since critical parameters may not have been reported and tracking down the required information such as site coordinates and the date of observation is increasingly difficult with greater amounts of time since studies have been published.

To offset the perception on an unintentional bias toward fractionation studies, we have added additional text in several locations in order to emphasis the inclusion of the other forms of radiocarbon constraints mentioned by Dr. Baisden. For example, we will modify the third paragraph of the introduction as follows:"The pool partitioning approach is easily implemented in SOM models, but in reality, measuring these pools is both challenging and dependent on the techniques used to fractionate the bulk soil (Moni et al., 2012) or to track throughput of bomb-derived carbon through repeat measurements (Baisden et al., 2013; Basiden and Keller, 2013)... Critically, approaches using radiocarbon to estimate the timescales of carbon cycling in soils require multiple measurements of carbon in distinct soil reservoirs (Trumbore, 2000) or through time (Baisden and Canessa 2013; Basiden and Keller, 2013)..

I provide additional detail and discussion in relation to particular lines in the manuscript.

L39-41 The sentence spanning these lines is problematic for several reasons. First, on the face of it, its assertion regarding bulk carbon appears to me to be disproven by Baisden et al (2001; 2013a, 2013b). This may simply be a matter of interpretation however – I also have trouble linking this sentence to what follows it, given what is

possible in quantifying the stabilisation and turnover of carbon. Second, the confusion I see in this sentence may perhaps lie in what is meant by the word, "predict." To play devil's advocate on this point, I'll suggest that far more would be known and quantified if, since 2000, the field had followed the simple process of collecting and running time series bulk samples, and using math to separately measure the size and turnover rates of pools. In contrast, it doesn't seem that ongoing efforts to chemically and/or physically fractionate soil have led to clarity or application.

Response: The reviewer's opinion on this is well-known, and in places where archived samples are available, can be a valid point. However, the models used to describe time series require creating organic matter 'pools' that cycle carbon at different rates. The use of physically and chemically separated fractions, while imperfect, is an attempt at linking model-derived turnover times with corresponding physical or chemical stabilization mechanisms. We will replace the sentence spanning lines 39-43 to better reflect this: "Bulk soil radiocarbon measurements, if not part of repeated time series, provide only a mean estimate of the time elapsed since carbon in the soil was fixed from the atmosphere. However, this mean is not representative of how fast soil C will respond to a change in inputs, as it has been repeatedly demonstrated that SOM is not homogeneous, and that C stabilized by different physical, chemical or biological mechanisms cycles at different rates. Models used to explain time series of bulk radiocarbon (e.g. Baisden et al (2001; 2013a, 2013b) or physically and chemically separated soil organic matter fractions (Gaudinski et al. 2000; Sierra et al. 2012, Schrumpf et al. 2013) require model structures with multiple pools cycling on different timescales to simultaneously explain the rate of bomb 14C uptake and the mean 14C signature of SOM."

L42 "mean ages and cycling rates" are duplicative, since rate is the inverse of age given simple pools. Also, why imply "mean" ages? Mean ages, especially when used across distinct pools, or without time series data imply considerable risks of biased results (Baisden et al 2013a) so I would like to see the community to be careful and

precise in the use of this type of terminology.

Response: We agree that the use of the term "age" in this sentence is confusing, especially in the context of radiocarbon, and will remove it to make it clearer that we imply partitioning of the SOM in pools of different cycling rates. We agree in that we need to use more precise definitions, particularly because the use of terms such as age, residence time, turnover time, cycling rate, etc. are sometimes used interchangeably leading to confusion. For this reason, we adopt here the more precise definitions of system age, radiocarbon-derived age, turnover time and transit time as defined in Sierra et al. (2017, Global Change Biology 23:1763).

L45 The introduction of transit time as a completely different measure is confusing. A great deal of work has included an understanding of transit, for example by explicitly attempting to model transport processes within soil. It may seem pedantic, but it quite important to understand that "transit" times are useful in systems where transport is important as a process. This distinction should either be left vague, noting that the measures differ somewhat, or be better expanded to recognise work focussed on transport. Useful examples include Elzein and Balesdent 1995, Baisden et al 2002; Baisden and Parfitt 2007, Sanderman et al 2008, and Jenkinson and Coleman 2008.

Response: We respectfully disagree that this measure is confusing, as transit time has a clear definition with a long history (Bolin & Rodhe 1973, Tellus 25:58) and has been used as a mathematically derivable characteristic for the kinds of compartmental models used to model the terrestrial carbon cycle globally (Fung et al. 1998 Global Biogeochemical Cycles; Metzler et al. 2018 PNAS) and in soils (Manzoni et al. 2009, J. Geophys Res 114, Sierra et al. 2018, Global Biogeochem. Cycles 32:1574). There is not necessarily a link between the transit time (a characteristic of the model and the conditions in which it is run) and the process involved (which can involve transport as well as a given stabilization mechanism). We think it is critically important to distinguish between system age (the age of carbon in the soil) versus transit time (the time it takes carbon to pass through the soil system, since the time it enters until it leaves as $CO_2$

or DOC). We explicitly avoid the most confusing term residence time because it has been used in the literature to imply either age or transit time. By making this distinction we expect to have a more specific interpretation of radiocarbon data measured in bulk soil and pools versus radiocarbon in respired CO2. Also, system age and transit time are more precise terms that address model-derived quantities more directly.

L48-49 This statement appears incorrect. It certainly has been shown mathematically tractable to separate distinct 'pools' without physically or chemically separating soil. For grassland soils, the comparisons in Baisden et al 2002, and further work in Baisden et al 2013 and 2013a make a fairly clear mathematical separation with time series samples is more efficient. Undoubtedly options may vary on this topic, but at a minimum the case for mathematical separation based on bulk samples has to be acknowledged as valid strategy. This is particularly true if total throughput of C through the ecosystems can be understood (Gaudinski et al 2000; Sierra et al 2012; Baisden and Keller 2013).

Response: This is an important point and we regret not addressing this topic in our original draft. We will add language to reflect the excellent work applying a mathematical separation of soil carbon pools, using a time series of measurements. However, we would point out that purely mathematical pools may describe the evolution of 14C, but do not necessarily help us to understand the underlying processes; for this a combination of physical and chemical separation methods that emphasize different kinds of stabilization mechanisms, combined with mathematical modeling.

L51-52 The references given for the shifting view of controls on soil organic matter dynamics give an unfair impression of recent progress, using papers that do present useful recent synthesis. It seems remiss not to include earlier references, or at least Oades 1989. It would be preferable to include Golchin et al 1996 as well.

Response: We agree and appreciate the suggestion to include citations to additional seminal publications. We will cite the studies listed.

L57-59 It might be more accurate to say there are either one or three things here, but

not two? If there is a 'fast' pool, and a slow 'pool' then different processes govern the turnover of each, so the two processes each need parameters. But it is equally important that the process of partitioning carbon flows into soil between the two pools be understood. Yet, I could also see another point of view, that there are typically more than two pools recognised in soil, so perhaps an understanding of partitioning only is intended here? Please clarify.

Response: We have clarified our list of questions driving research of SOM dynamics by adding an additional question to our list:"Three of the fundamental questions currently driving SOM research are: (1) What are the controls on the partitioning of organic inputs between soil reservoirs cycling over different timescales; (2) what factors determine the fraction of organic inputs to soil that are lost, retained, or transferred each reservoir; and (3) which mechanisms contribute to the stabilization or protection of SOM?"

L 68 It seems slightly odd not to have pioneering or earlier exemplars of density fraction in this list. Various students of Oades, and particular series of papers published in 1995-7 by Golchin. Keep in mind that many of these methods were not developed specifically for radiocarbon.

Response: This is a very good point, many of or reference have focused on the application of radiocarbon measurements to fractionation methods, however, such methods have often been developed and applied in the context of other soil measurements. As per the reviewer's suggestion, we will also cite the early work of Golchin et al. in developing and interpreting such methodologies.

L84 It is interesting here to see version 1.0 (Sierra et al 2012) rather than version 1.1 (Sierra et al 2014) of SoilR referenced. Please see the note below regarding L102 about an interface to soilR. If the goal of the database is to allow improved testing of hypotheses representing understanding of soil carbon dynamics, it seems SoilR should provide an ideal mechanism for implementation. It would be good to see more clarity

of thought on achieving this, including a reference to the later version of SoilR.

Response: As requested, we will an updated reference to SoilR version 1.1 by Sierra et al. 2014. See also our response to additional comments regarding linkages to SoilR below.

L88-89 It may be worthwhile considering earlier references to DOM such as Sanderman et al 2008. I say this in part, because what is said in this paper may guide the use of the database, and it would be worrisome to neglect early studies containing compelling radiocarbon results.

Response: We will add a reference to Sanderman et al., 2008, as requested.

L102 Here again, I'd propose there is a collective forgetfulness of what was well established in the literature by the 1990s in terms of paradigms of soil organic matter dynamics. These have been reinforced by review and synthesis in recent decades, but this is not a reason to neglect early radiocarbon work that had already largely incorporated the paradigms promoted in this introduction. Therefore, it is odd to see early work that established overall constraint of carbon dynamics in well-studied systems neglected here. Such work can provide useful examples of how to construct strategies for constraining carbon dynamics. The obvious examples driven entirely by radiocarbon are Gaudinski et al 2000 (in relation to followup by Sierra et al 2012) and the set of work in Baisden et al 2002, 2002a, 2003.

Response: Dr. Baisden makes an excellent point that there is long history of research supporting our current paradigms for soil organic matter dynamics. While this manuscript is not intended to be a comprehensive review of this research area, it is important for us to honor the diversity and breadth of the work underlying the database. As such, we will make an effort to incorporate the majority of citations suggested by Dr. Baisden.

A second issue the lack of reference to or inclusion of literature using tracer carbon, or

natural abundance stable C isotope ratios.

Response: We have made a conscious decision to exclude radiocarbon values from tracer studies in this version of the database. This is something that we hope to accommodate in the future but at this time we feel that the potential for problems from inadvertently merging natural abundance and tracer data outweigh the benefits of including them.

With regards to natural abundance stable C and N isotopes, these are included in the database and we encourage submission of studies including such observations. We will add an additional statement to the manuscript to reinforce these points:"The IS-RaD (v1.0) is designed to be an open-source platform that (1) provides a repository for soil radiocarbon and associated measurements, (2) is able to accommodate data collected from a large variety soil radiocarbon studies, including the diversity of fractionation techniques applied to soils as well as repeated bulk measurements made over spatial or temporal gradients, and (3) is flexible and adaptable enough to accommodate new variables and data types. Although ISRaD was specifically developed with soil radiocarbon measurements in mind, it is well suited for synthesizing other soil measurements, including stable C and N isotopes. Importantly, we currently focus only on natural abundance isotopic measurements and therefore exclude data from isotopic tracer studies."

Finally, a weakness in papers on recent paradigms is the importance of closing the partitioning and turnover of soil carbon by constraining the overall flow of C through the system via NPP or respiration. This is a strength of SoilR (Sierra et al 2014) so, as noted above, I would like to encourage the authors to consider what link might be made between these two R packages. This is covered to some degree in L225-231 but not with explanation of the value of or rationale for such constraints.

Response: We agree that there are fruitful linkages to be made between ISRaD and SoilR. Such collaborations have been discussed and, in some cases, may already be

underway. However, our first priority has been to formalize and finalize the database. Future users of the database need to recognize that any quantitative interpretation of soil C dynamics using radiocarbon will use some kind of model that links radiocarbon to rates of C transfer or decomposition. While SoilR is a useful platform because it has made it easy for users to incorporate radiocarbon into most major soil C models in current use, the choice of model and platform should be up to the users of the database. Since both ISRaD and SoilR are implemented in the R language, users may find it convenient to use both packages in specific analyses, but we are not prescriptive about how these analyses should be done.

L131 It would be good to clarify here that the site accessible via the soilradiocarbon.org address does not appear to have an R-shiny interface or some other "web interface" to the data running. Either the words "web interface" should be changed to "web site", or an exact address to a "web interface" should be provided.

Response: We agree that our original terminology is confusing. We intended the term "web interface" to describe a browser-based access point for the database and we did at one time have access to an R-shiny interface from the web site. However, technical limitations that ultimately led to us abandon that approach (the R-shiny interface is still available in the R-package). So, in fact, as suggested, the "web interface" is really just a "website." As per the reviewer's suggestion, we will change our wording throughout the manuscript.

L270-279 It is good to see these items related to density fractionation included specifically. However, does it make sense to include/explain these stored values but not include the degree to which the sonication method has enabled isolation of occluded vs free light fraction, again originally detailed by the Golchin work to adapt density fractionation to the paradigms the authors promoted at the beginning of this manuscript's introduction?

Response: This is a great suggestion and one that we have discussed. Our approach

to the categorization of various fractionation approaches has been to capture all relevant information required to compare measurements across diverse datasets with as few required variables as possible. Sonication details are one aspect that we chose not to include. Instead, we report whether sonification was performed but not the specifics of the energy used or the time over which it was applied. In many cases, this information is often not reported in the original studies. That said, we agree that this can be useful information that can be included in comment fields that are available in the template where data are entered.

We have a process in place for data users to request the inclusion of new variables (described in section 4.3). The simplest way to accomplish this is the post an issue in the GitHub repository but users may also email info@soilradiocarbon.org and an issue will be posted to the GitHub repository for them. Once an issue has been posted, there may be some discussion amongst other users as to if the requested variable warrants inclusion and how best to implement its addition. Pending the results of this discussion, the science steering committee will approve or reject the inclusion of the variable. If it is approved, the technical changes required to implement the change will be added to the queue for coding changes. When requested variable additions are not intended to be mandatory, their incorporation is straightforward. When they are suggested as a mandatory variable, the implementation is more difficult and an effort must be taken to "back-fill" existing datasets, wherever possible. Such efforts will require time commitments from users to complete effectively.

To address this suggestion by Dr. Baisden, we have created a new issue in the GitHub repository: https://github.com/International-Soil-Radiocarbon-Database/ISRaD/issues/199

L320 The web interface again appears to be a regular website rather than a web database interface?

Response: We will change this terminology as requested.

L322 This web address only goes through with http:// and not with https:// L332 The ISRaD package was not available at cran.org as implied in this text. The Github version indicates changes. Although these changes are probably minor I was disappointed to find that there was not a version tagged to support this review process.

Response: As describe above, we have made changes to how the R-package is structured in order to allow us to meet the requirements of CRAN and plan to have the R-package available as soon as possible. We would also like to note that, while it is unfortunate that the R-package was temporarily unavailable, our multitiered approach for serving the data (i.e., direct download from the website, access via the R-package, download to active and archived versions GitHub, or download of archived versions via the Zenodo) do provide redundancies so that data can still be accessed if technical issues are encountered in one or more of the access points.

L501 Here again the "web interface" is mentioned. It seems worth noting here that this link appears to lead to a fairly standard website with a static file download for a database table, rather than an interface to the database. What's missing? There seems to be neither an accounting of the spread of categories or types the data already in the database represent, or what weaknesses (gaps) can be described. Similarly, there is a rather technical description of data entry, but not a description of how substantial historic datasets might be brought into the database. An additional but admittedly problematic question is whether the extent of available published data not in the database can be better quantified and described. I encourage some discussion of these opportunities for improvement.

Response: We acknowledge that the "web interface" does not provide some of the pertinent details regarding the database mentioned. However, our feeling is that this manuscript will serve to fill that need initially and, pending is publication is ESSD, we will provide links to this document from the website. It is our opinion that the open-review process provided by ESSD, provides a better forum for initially describing the details mentioned by Dr. Baisden.

Additionally, in regards to the point made about the spread of data in the database, we have built a few simple querying and reporting functions in R that facilitate the assessment of the number of studies or the number of data records for specific variables. For those users familiar with R, it is a relatively simple matter to use standard R functions to query the ISRaD data object.

Furthermore, on the ISRaD website we host a collaborative document listing the studies that are currently in progress as well a "wish-list" of studies that we intend to ingest in the future.

---

## Author Response (AR2)

Author's Response to Reviews
essd-2019-55

We would like to again thank the editor and reviewers for their thoughtful comments, which have greatly improved this manuscript. In our latest revision, we have addressed the additional comments from Dr. Troy Baisden. His suggestions are pasted below (in black) with our responses shown in red.

**Reviewer (Troy Baisden) Comments on Revision 1**
Overall, the response to review and revised manuscript is in good shape and I'm very pleased with the revised manuscript. I appreciated the constructive and generally well written responses to the review. I reiterate my view that this publication is the culmination of an extremely valuable piece of work, albeit one that probably became bigger than the authors imagined. It's completion is a considerable achievement, and although there are no guarantees, it has real potential to enable considerable progress in the field. I spotted some remaining quibbles (typos, lack of clarity in references, etc) which I outline at the end of this review, and somewhat of a lack of clarity in the additional key question introduced (where a typo or omitted word makes it hard to interpret meaning). And last, there is a substantive difference on the semantics of age, and transit vs residence times and alternate words for the same concepts. I've outlined some clear thoughts on this issue below, but the net effect is that the manuscript should be publishable without the need for further discussion if authors ensure the final version simply encourages users to be aware of the need to consider the concepts of age, transit, and residence or turnover times and the related definitions and assumptions in using IsRad.

*Semantics of Age, Transit Time and Residence Time*
In response to my comments noted against Line 45, I think there is necessarily a bit of an impasse regarding the use of "transit" time. It has minor implications here, and is an editorial issue in terms of communicating to a wide audience (e.g. across disciplines). Let's just say there remains a "muddle" that has been very usefully identified but not resolved (in my view) by work involving the authors, particularly Sierra et al 2017 (which is published as an "Opinion" article and not as a review, and in the journal Global Change Biology which is not likely to get good review representing the less 'biotic' of the earth system sciences).

The best solution for the publication of this work, explained in detail below, seems to be to use accessible language and/or suggest that more effort is needed to clarify workable definitions across disciplines. Given the purpose of this paper, I see no need to reach a forced resolution, but will try to clarify my view for the editor and authors given the response to the review comments.

The heart of my concern is that the technical language we use for specialised research should remain consistent with standard dictionaries and textbooks in the relevant (connected) disciplines. And, we should pay attention to it to the degree that misinterpretation or misunderstanding can cause severe errors. Starting with the dictionary, transit is strictly an adjective here, but normally defined as a noun: 1) carrying of people or things to one place or another; 2) the action of passing through a place. A verb definition is usually linked to (2). Similarly, "residence" has the relevant definition (noun): the fact of living in a particular place, and related concept of 'in residence'. In the response to reviews, the authors start with Bolin & Rodhe 1973, who carefully place residence time in parentheses after each introduction of transit time. Transit time is clearly meaningful and specific directly in relation to its dictionary definition (passing from one place to another) across areas of the earth system sciences including hydrology/hydrogeology, geomorphology and oceanography.

I think success within papers like Bolin and Rodhe as well as the others the authors point to in their response is that it can be insightful to conflate the concepts of residence times and transit times, where the boundaries of the system can be placed so that transport processes can be simplified or removed from consideration. I still think this is best considered a special case, but is definition dependent. The problem with generalising this equivalence may be that any such simplification relies on assumptions and understanding, which if ignored (and untrue) causes errors. These errors are most likely to occur and be compounded when there is a mismatch between specialised terminology and plain english, and it is an important (and sorrowfully annoying) role of reviewers and editors to try to prevent this, without carrying out a full modelling analysis or a review across several disciplines.

Author's Response to Reviews
essd-2019-55

In their response, the authors suggest that 'residence time' has been conflated with both age and transit time, but I think that is simply the result of it being the most commonly used term. It is in fact well introduced to many students in text books (see below). In addition, it may be useful to point to Maloszewski and Zuber (1982 J Hydrol 57:207) as an well-cited example where the conflation of all three (age, residence time and transit time) is clearly stated, and can be linked to the causation of significant problems over the years in applications. The problems with this particular work are relevant here because they have occurred in the interpretation of post-bomb tritium results, including time series, which has similar mathematical dynamics to soil radiocarbon. In my view, residence time can be usefully considered the same as transit time, but may also be useful to distinguish at different scales or in models where transport between reservoirs can be removed or collapsed (see Baisden et al 2013).

In this work, it would be ideal to maintain consistency with language used in standard/classic textbooks (which regularly define "residence time") in fields such as environmental modelling (Harte), ecosystem ecology (Odum), limnology (Vincent), isotope geology (Faure), biogeochemistry (Schlesinger) and geochemistry/geochemical modelling (Albarede). I have checked my collection of books in these fields (authors noted in the previous sentence), and none of these index transit time as key concept. If a more unusual term, such as transit time is used, and claimed to provide insights or improved definitions, it still needs to be defined – which is not done in the revised manuscript. Oblique citations are not enough. Given prevalence in textbooks in relevant disciplines, the unexplained but strongly worded decision by the authors to avoid the use of the widely used 'residence time' concept is problematic, and seemingly circular if they return to Bolin and Rodhe 1973 to explain this. Ultimately, there probably isn't reason in this paper to discuss or deal with where the terminology fails or causes confusion around areas such as non-steady state systems, and just a need to define rationale and future use of the IsRAD. Many of the authors target better understanding of transport as a component of the C cycle as a key goal, both down through profiles and across landscapes/catchments. The latter is increasingly recognised as important in the carbon cycle on the timescale at which radiocarbon can be used to constrain dynamics. As a result, it would be good to be careful to avoid "muddle" and potential bias, and retain options for further innovation and clarification in terminology associated with these terms.

So for this paper, I don't see much to do except be mindful that the authors and I agree that there is an important problem of semantics to be aware of, but because I am not convinced of their solution, the problem can be framed minimally within the paper so that use of the database can help the field move toward consistent solutions to these semantic issues – ultimately clearly quantifying and communicating the dynamics of systems. I provide some suggestions for slight changes or notes to accept the text as is below.

**Author Response**

We appreciate the effort of Dr. Baisden in helping to clarify the terminology issues that arise when trying to disentangle different aspects of the timescales at which carbon is stored and cycled in soils.

We agree in that the term 'residence time' has been widely used in the past to infer something about the dynamics and timescales of carbon cycling. However, when the problem is analyzed in more detail, it is clear that one needs to differentiate between the age of carbon stored in the system versus the age of carbon in the output flux. We chose calling the later 'transit time' because there is a history of attempts to disentangle these concepts, trying to avoid the more ambiguous term 'residence time' that could be interpreted as both.

Following Dr. Baisden suggestion, we provide new definitions of these terms in the manuscript and equate transit time to residence time. We provide the following definitions:

Carbon age: is the age of carbon stored in the soil, since the time it enters until a time of observation.

Transit (residence) time: is the time that carbon needs to pass through the soil system, since the time it enters until it is observed in the output flux. This output flux can be defined as the respired $CO_2$ flux leaving the soil or the amount of DOC in the runoff flux.

Author's Response to Reviews
essd-2019-55

We hope these definitions not only help to better clarify the concepts introduced in the manuscript, but also help to better understand the different types of data provided in the database.

**Specific points**
L2 remove 'the' at end of line? There is odd syntax resulting from preceding terrestrial ecosystem dynamics and not parallel to use of dynamics earlier in sentence, implying dynamics of two related systems.
We have made the suggested change.

L22 Use of terms describing dynamics seems fine here, but compare to L 106.
We left the terminology as stands here but made the suggested changes at L106.

L52 There is no 2001 publication by Baisden or Baisden et al. 2001. I apologise - this may relate back to a typo in my review. Presumably Baisden et al 2002a is intended, although this would now feel slightly odd when placed here because it contains a comparison of methodologies including both bulk and fractionation. (Most of the fractionation is in Baisden et al 2002b, but specialised fractionation below the A horizon is only in 2002a.) Overall, I suggest perhaps reconsidering how this sentence is split and referenced, but think the message here is about right. It might be simple to move reference to the end? The only caveat is that the inclusion of 'synthetic constraint' (resulting from the inclusion of NPP or related measures to constrain total C turnover from all pools) could be useful as defined/promoted by Baisden and Keller 2013, but also applied in Baisden et al 2002a (and reexamined in Baisden and Parfitt 2007)
We thank Dr. Baisden for pointing out this discrepancy, we have modified our citations of his work based on this suggestions.

L58 and elsewhere Baisden is spelled wrong in "Basiden" and Keller…
We apologize for this spelling error, which occurred in our reference tracking software and, as a result, was propagated throughout the manuscript. We have corrected all instances of this mistake.

L63 Is "transit" really intended here? Can residence time(s) be added in parenthesis as in Bolin and Rodhe as noted in the discussion above. Or 'turnover times' because this is clearer when applied to separate fractions?
As per Dr. Baisden's suggestions we have added additional clarification for our definitions and also added "residence" in parenthesis.

L 77 There's a noticeable typo where I can't figure out what was intended in the middle of the now three questions "2) what factors determine the fraction of organic inputs to soil that are lost, retained, or transferred each reservoir;" I might suggest rephrasing this to better pick up on all four main processes (additions, removals, transfers, transformations) recognised traditional in pedology and extended to ecosystem science across the suite of three questions. If so, question 1 describes inputs and the partitioning of inputs, "what factors determine rates at which SOM in each reservoir is lost, or retained, or transferred or transformed within the soil?" This approach leaves the third question as a part of transformations (to more protected forms).
We have modified these questions for clarity and to address the four main pedological processes reference by Dr. Baisden. The questions now read as:

*"…(1) what are the controls on the partitioning of organic inputs between soil reservoirs cycling over different timescales; (2) what factors determine rates at which SOM in each reservoir is lost, retained, or transferred within the soil; and (3) which mechanisms contribute to transformation of SOM to stabilized or more protected forms?"*

L96 Now that Golchin et al 1994 is cited, I'd suggest referencing it here with regard to sonication and occluded fractions?
We have added this reference to the sentence describing sonication and occluded fractions, as suggested.

L106 I don't strongly disagree with what is said here, but the insight provided by the response to review suggests that the following text "as well as differentiating between the mean age and the transit time for the whole soil"

might be usefully changed to "as well as differentiating between different measures of dynamics ranging from mean age to the transit time for the whole soil"
We have incorporated this suggested change.

As a last point, I don't have a specific location to recommend, but I do note that giving the authors an opportunity to consider inserting or further emphasising a sentence or two somewhere outline the reasons why the definitions and assumptions that cause difficulty in the interpretation of the concepts of age, transit time, residence time etc deserve focus for users of IsRad.
We agree with Dr. Baisden that this is an important point to emphasize. We have added the following sentence: "*As the assumptions required for modeling radiocarbon data can lead to confusion in the terminology and concepts of SOM dynamics, it is imperative that we archive radiocarbon measurements in order to preserve the ability to reevaluate calculations and compare data across different modeling frameworks.*" To the end of the 3rd paragraph to highlight how the database is an important tool for addressing confusion or disagreement in concepts of SOM dynamics.